# Seasonal Arctic sea ice forecasting with probabilistic deep learning

Tom R. Andersson [1]✉, J. Scott Hosking [1,2], María Pérez-Ortiz[3], Brooks Paige[2,3], Andrew Elliott [2,4], Chris Russell [5], Stephen Law [2,6], Daniel C. Jones [1], Jeremy Wilkinson[1], Tony Phillips[1], James Byrne [1], Steffen Tietsche [7], Beena Balan Sarojini [7], Eduardo Blanchard-Wrigglesworth[8], Yevgeny Aksenov [9], Rod Downie[10] & Emily Shuckburgh[1,11]

Anthropogenic warming has led to an unprecedented year-round reduction in Arctic sea ice extent. This has far-reaching consequences for indigenous and local communities, polar ecosystems, and global climate, motivating the need for accurate seasonal sea ice forecasts. While physics-based dynamical models can successfully forecast sea ice concentration several weeks ahead, they struggle to outperform simple statistical benchmarks at longer lead times. We present a probabilistic, deep learning sea ice forecasting system, IceNet. The system has been trained on climate simulations and observational data to forecast the next 6 months of monthly-averaged sea ice concentration maps. We show that IceNet advances the range of accurate sea ice forecasts, outperforming a state-of-the-art dynamical model in seasonal forecasts of summer sea ice, particularly for extreme sea ice events. This step-change in sea ice forecasting ability brings us closer to conservation tools that mitigate risks associated with rapid sea ice loss.

[1] British Antarctic Survey, NERC, UKRI, Cambridge, UK. [2] The Alan Turing Institute, London, UK. [3] Department of Computer Science, University College London, London, UK. [4] School of Mathematics and Statistics, University of Glasgow, Glasgow, UK. [5] Amazon Web Services, Tübingen, Germany. [6] Department of Geography, University College London, London, UK. [7] European Centre for Medium-Range Weather Forecasts (ECMWF), Reading, UK. [8] Department of Atmospheric Sciences, University of Washington, Seattle, WA, USA. [9] National Oceanography Centre, Southampton, UK. [10] WWF, Woking, UK. [11] University of Cambridge, Cambridge, UK. ✉email: tomand@bas.ac.uk

Near-surface air temperatures in the Arctic have increased at two to three times the rate of the global average, a phenomenon known as 'Arctic amplification', caused by a number of positive feedbacks[1–3]. Rising temperatures have played a key role in reducing Arctic sea ice, with September sea ice extent now around half that of 1979 when satellite measurements of the Arctic began[4]. This downward trend will continue, even in optimistic greenhouse gas emission reduction scenarios[5]. Climate simulations project the Arctic to be ice free in the summer by 2050[6]. Other studies put this date as early as the 2030s[7]. Such unprecedented sea ice loss has profound local and regional consequences: it is the greatest threat to polar bear populations[8]; it has increased the intensity and frequency of algal blooms that propagate toxins throughout the food web[9]; and it poses significant challenges for Indigenous Peoples, with impacts ranging from food security[9] to loss of culture[10].

Arctic sea ice is also a crucial component of the global climate system. Evidence is mounting that Arctic sea ice loss influences weather and climate beyond the Arctic region. For example, it may provoke wetter European summers through a southerly perturbation of the jet stream[11], as well as extreme Northern Hemisphere winters by weakening the stratospheric polar vortex[12,13]. Although the existence of such teleconnections are still in debate[14], improved forecasts of Arctic sea ice have the potential to improve predictions of mid-latitude weather[15].

Producing accurate Arctic sea ice forecasting systems has been a major scientific effort with fundamental challenges at play. Current operational sea ice forecasting systems, based on deterministic coupled atmosphere-ice-ocean models, are often no better than simple statistical forecasts at seasonal lead times of two months and beyond[16,17]. While there are inherent sea ice predictability limits, owing mostly to chaotic processes in the atmosphere[18–20], studies have demonstrated that potential predictability is higher, suggesting that forecasts could be improved[17,21,22].

In this work, we introduce IceNet, a new sea ice prediction system based on deep learning. The system has been trained to forecast the next 6 months of monthly averaged sea ice concentration maps at 25 km resolution, learning from climate simulations covering 1850–2100 and observational data from 1979 to 2011. We show that IceNet outperforms a leading physics-based model in seasonal forecasts of Arctic sea ice,

particularly for extreme summer sea ice events. IceNet directly predicts probabilities of sea ice occurring, expressing the level of confidence in its own predictions, unlike previous deterministic models. We find that IceNet's predicted probabilities of sea ice display good calibration with observations. Leveraging this, we derive a simple framework for probabilistically bounding the ice edge within a region of lower predictive confidence, which has added utility over deterministic ice edge forecasts. Finally, a variable importance method is used to identify the climate variables most important for IceNet's forecasts.

## Results

**IceNet: a sea ice forecasting AI.** In contrast to physics-based dynamical models are data-driven artificial intelligence (AI) approaches like deep learning. Deep learning algorithms have been a game-changer in diverse areas where large volumes of data are available, using multiple nonlinear processing layers to extract increasingly high-level information from unprocessed input data[23]. There is great interest in the application of deep learning to the Earth sciences[24], particularly with satellite data[25]. Satellite and climate model data are gridded; a specific time and altitude slice of a climate variable is arranged on a two-dimensional $(x, y)$ grid, analogous to an image, and can be used as inputs to convolutional neural networks (CNNs)[26–28]. Satellite observations of sea ice are also presented as images: passive microwave measurements of microwave brightness temperature are converted to sea ice concentration (SIC) estimates of the fractional area covered by sea ice in a given grid cell, ranging between 0 and 100%.

Given the applicability of CNNs to satellite and climate model data, we designed IceNet as an ensemble of U-Net networks (Fig. 1 and Supplementary Table 1). The U-Net, initially developed for medical imaging segmentation[29], is a CNN variant that takes images as input and produces images as output, and has proven effective in diverse applications at learning accurate, pixel-wise mappings[30,31] (see Methods). IceNet's monthly averaged inputs comprise SIC, 11 climate variables, statistical SIC forecasts, and metadata (Supplementary Table 2), stacked in an identical manner to the RGB channels of a traditional image, amounting to 50 channels in total. Each IceNet ensemble member is trained to predict the future six months of monthly averaged SIC maps.

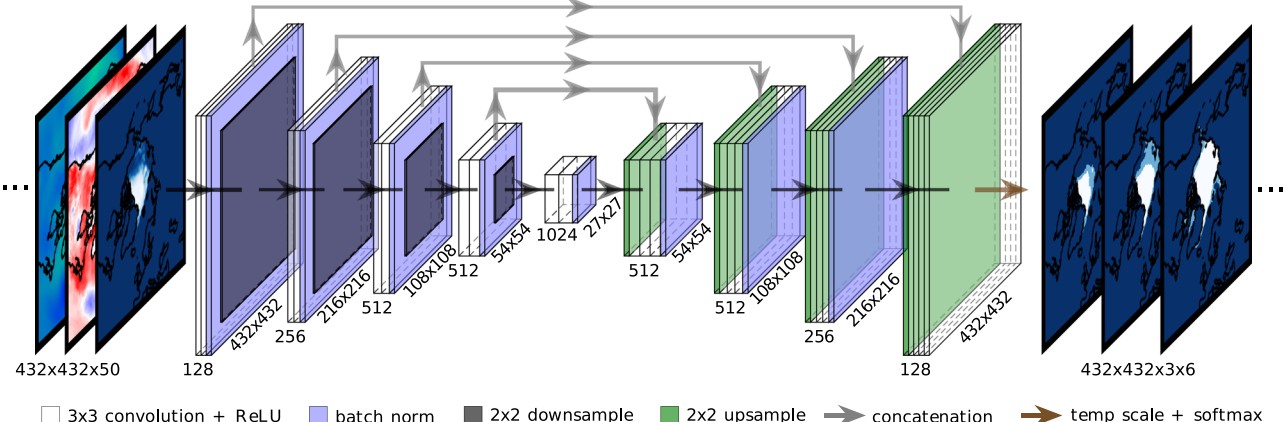

**Fig. 1 The IceNet model.** IceNet receives 50 monthly averaged climate variables as input (Supplementary Table 2), centred on the North Pole. IceNet, a deep learning U-Net model, receives these inputs and processes them through a series of convolutional blocks with batch normalisation (Supplementary Table 1). The number to the left of the convolutional blocks denotes the number of feature maps in each convolutional layer, while the number beneath denotes the feature map resolution. IceNet's outputs are forecasts of three sea ice concentration (SIC) classes (SIC ≤ 15%, 15% < SIC < 80%, and SIC ≥ 80%) for the following 6 months in the form of discrete probability distributions at each grid cell. The latter two ice class probabilities are summed to obtain the sea ice probability, $p = P(\text{SIC} > 15\%)$.

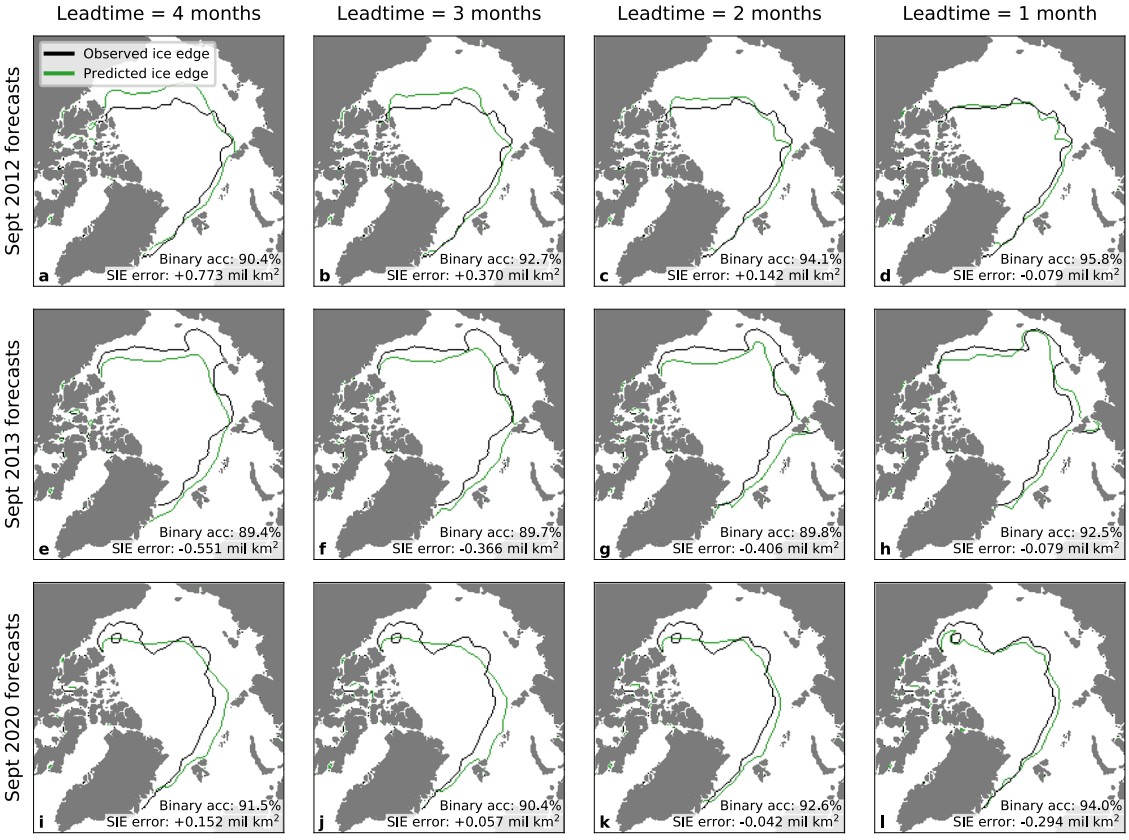

**Fig. 2 IceNet's ice edge forecasts for extreme September sea ice events at 4- to 1-month lead times.** Forecasts are shown for September 2012 (lowest ice extent on record) (**a–d**), September 2013 (anomalously high ice extent) (**e–h**), and September 2020 (second-lowest ice extent) (**i–l**). The observed ice edge (in black) is defined as the sea ice concentration (SIC) = 15% contour. IceNet's predicted ice edge (in green) is determined from its sea ice probability forecast as the $P(\text{SIC} > 15\%) = 0.5$ contour. The binary classification accuracy and sea ice extent (SIE) error is shown for each forecast (see 'Evaluation of IceNet's performance' section). 2012 and 2013 are in IceNet's validation dataset and 2020 is in its test dataset.

Past studies have used deep learning for SIC prediction to some success, such as a single grid cell-wise neural network[32] and a sliding window CNN[33]. Both of these approaches limit the input receptive field and thus the scale of spatial interactions that can be modelled. Owing to the U-Net architecture used for IceNet, each grid cell's forecast receives information from over 1500 km in the $x$ and $y$ directions of the input data, enabling long-range spatiotemporal interactions to be modelled.

To reduce the effect of uncertainty in the SIC data[34], the problem is framed as a classification task with the output SIC values divided into three classes: open-water (SIC ≤ 15%); marginal ice (15% < SIC < 80%), and full ice (SIC ≥ 80%). At each grid cell and lead time, IceNet's ensemble members produce a discrete probability distribution over each of the three SIC classes. IceNet's ensemble-mean output is found by averaging the individual probability distributions of its 25 ensemble members (see Methods), which improves performance and probability calibration[35,36].

The marginal ice class was included to increase the expressivity of IceNet's forecasts. However, to simplify model evaluation, IceNet's marginal ice and full ice class probabilities were summed to obtain $P(\text{SIC} > 15\%)$, hereafter referred to as the sea ice probability (SIP), $p$, with binary classes open-water (SIC ≤ 15%) and ice (SIC > 15%). This aligns with previous work: 15% is the standard SIC threshold for defining the ice edge position[37,38]. By reducing the task to binary classification of SIC > 15%, the objective can be framed as that of predicting the ice edge. Examples of IceNet's ice edge predictions for September forecasts at 4- to 1-month lead times are shown in

Fig. 2, highlighting three anomalous events in the sea ice record. This shows how IceNet updates its forecasts using new initial conditions as the lead time decreases, with the predicted ice edge approaching the true ice edge.

To account for the limited observational data record, which spans only 41 years, we use transfer learning by pre-training each IceNet ensemble member on 2220 years of climate simulation data from the Coupled Model Intercomparison Project phase 6 (CMIP6)[39], covering the period 1850–2100. Each climate simulation includes anthropogenic forcing effects from greenhouse gas emissions, following emission levels since 1850 and projecting a 'middle of the road' scenario for the twenty-first century[40]. This scenario involves moderate shifts from fossil fuel to renewable energy sources, resulting in a net global average radiative forcing effect of 4.5 $\text{Wm}^{-2}$. After pre-training, systematic errors learned from the CMIP6 models are corrected by fine-tuning network weights on observational data from 1979 to 2011, followed by a probability calibration step known as temperature scaling[41]. The validation years from 2012–2017 were used for early stopping, hyperparameter search, and probability calibration, but were not used for training the models (see Methods). To further validate the predictive abilities of IceNet, test years spanning Jan 2018–Sept 2020 were left unused until IceNet was finalised, thus representing IceNet's true ability to generalise to unseen future data.

**IceNet's input variables.** Sea ice is dynamically and thermodynamically coupled to the atmosphere above and ocean below[42]. IceNet's 11 input climate variables (Supplementary Table 2) were

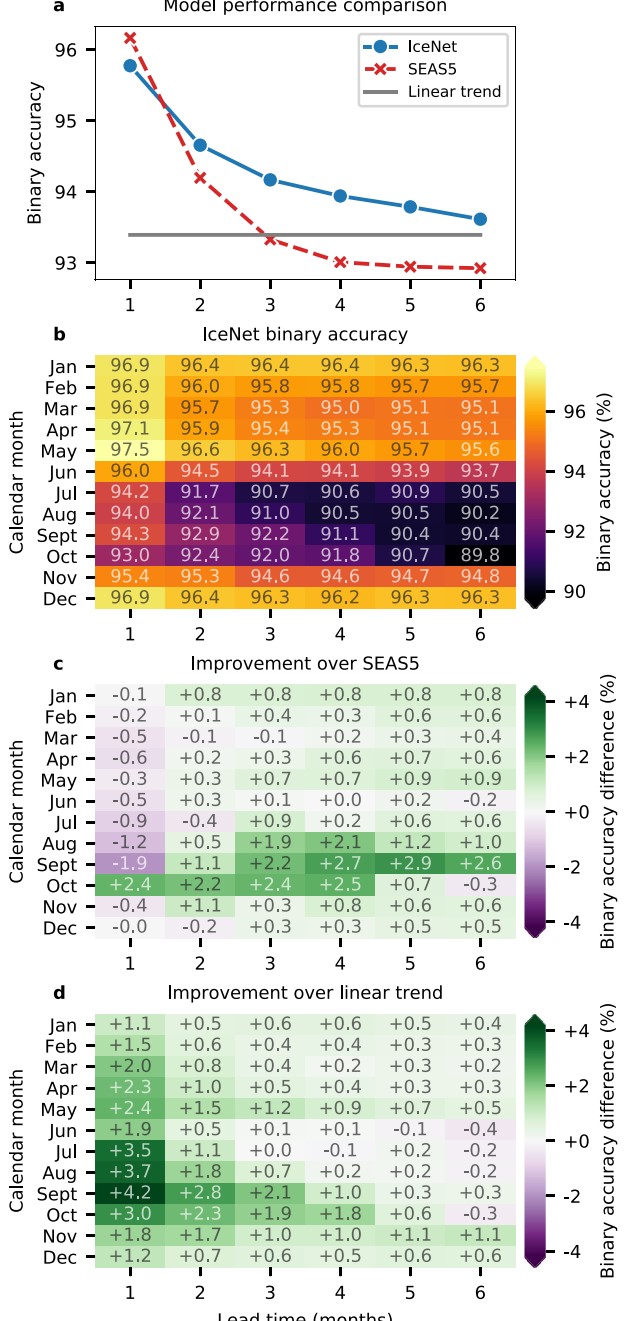

**Fig. 3 Comparing IceNet with dynamical and statistical prediction benchmarks. a** Mean binary accuracy versus lead time over the validation and test years (2012–2020) shown for IceNet, SEAS5, and the linear trend model. **b** IceNet's binary accuracy averaged across the validation and test years, shown for each forecast calendar month and lead time, with the heatmap values shown within each grid cell. **c, d** Heatmaps of the difference between **b** and the equivalent heatmaps of SEAS5 and the linear trend model respectively, illustrating IceNet's improvement over those models.

chosen to capture some of the principal mechanisms of such couplings. Sea ice melt or growth is driven by an energy balance from incoming radiation, as well as atmospheric and oceanic heat, necessitating the input of various temperature and radiation variables. Wind is a key driver for sea ice drift[43] so we chose to input near-surface wind, as well as geopotential height at 500 and 250 hPa to capture large scale circulation in the troposphere. Zonal wind at 10 hPa is input to IceNet to account for possible

teleconnections between the stratospheric polar vortex and negative anomalies in Arctic sea ice extent[44]. The initial state of the sea ice pack is also a key predictor, with persistence of sea ice anomalies potentially lasting seasonal timescales[21].

Not all variables relevant to changes in sea ice are sufficiently observed to be used in IceNet. For example, waves can break up the ice pack and ocean currents can move it around, but these fields are sparsely observed and therefore poorly constrained, so we chose not to include them. Furthermore, the formation of melt ponds on the ice pack in spring can also be an important driver of summer sea ice conditions[45], but there is no consistent pan-Arctic melt pond dataset over the 1979–2020 study period.

**Evaluation of IceNet's performance**. We compare IceNet with SEAS5[46], a dynamical model from the European Centre for Medium-Range Weather Forecasts (ECMWF) with state-of-the-art sea ice prediction skill[16,37,38]. For a fair comparison between the two, we use 25 ensemble members in both models (see Methods). As a statistical benchmark we use a SIC linear trend forecast, which extrapolates grid cell-wise lines of best fit— computed over the past 35 years of SIC values for a given calendar month—1 year ahead. Forecast performance relative to this benchmark represents an ability to forecast the interannual variations of sea ice beyond the linear decline component. SIC forecasts from the linear trend model are also passed into IceNet as inputs. This provides an additional layer of interpretability: the magnitude with which IceNet outperforms the linear trend model indicates how much IceNet can leverage its other input variables to forecast the nonlinear variations in sea ice under different forecasting regimes.

IceNet's probabilistic SIP outputs are mapped to binary class predictions of ice if $p > 0.5$ and open-water if $p \leq 0.5$. Deterministic forecasts of SIC in SEAS5 and the linear trend model were also converted to binary class predictions with ice if $SIC > 15\%$. Predictive skill was quantified using a binary accuracy metric, measuring the percentage of predicted SIC classes that match the observed SIC class. The binary accuracy is computed over an active grid cell region for a given calendar month and can be seen as a normalised version of the integrated ice edge error[47] (see Methods).

IceNet's mean binary accuracy across all lead times is only 0.13% higher on the validation than test years, suggesting its performance on validation data is also indicative of generalisation ability. Figure 3a shows the mean binary accuracy versus lead time over the 105 validation and test months for the three models, with IceNet outperforming SEAS5 and the linear trend model at lead times of 2 months and beyond. A heat map of IceNet's binary accuracy against calendar month and lead time (Fig. 3b) reveals the seasonal dependence of its predictive skill. IceNet extends the range of accurate forecasts, exceeding state-of-the-art performance at 2- to 4-month lead time forecasts for August, September and October, substantially outperforming both SEAS5 and the linear trend (Fig. 3c, d). IceNet's binary accuracy for 3-month September forecasts is greater than its benchmarks in 7 of the 9 held-out years for SEAS5 and in 8 of the 9 held-out years for the linear trend (Fig. 4 and Supplementary Fig. 1). SEAS5 generally outperforms IceNet at a 1-month lead time, though this is likely because IceNet only receives monthly averages as input, smearing the weather phenomena and initial conditions that dominate predictability on such short timescales.

IceNet's drop in predictive skill for long-range forecasts of summer in Fig. 3b reflects the 'spring predictability barrier', which affects all sea ice forecasting models. This predictability barrier arises primarily due to the importance of melt-season ice thickness for summer ice conditions[48]. Despite this, IceNet

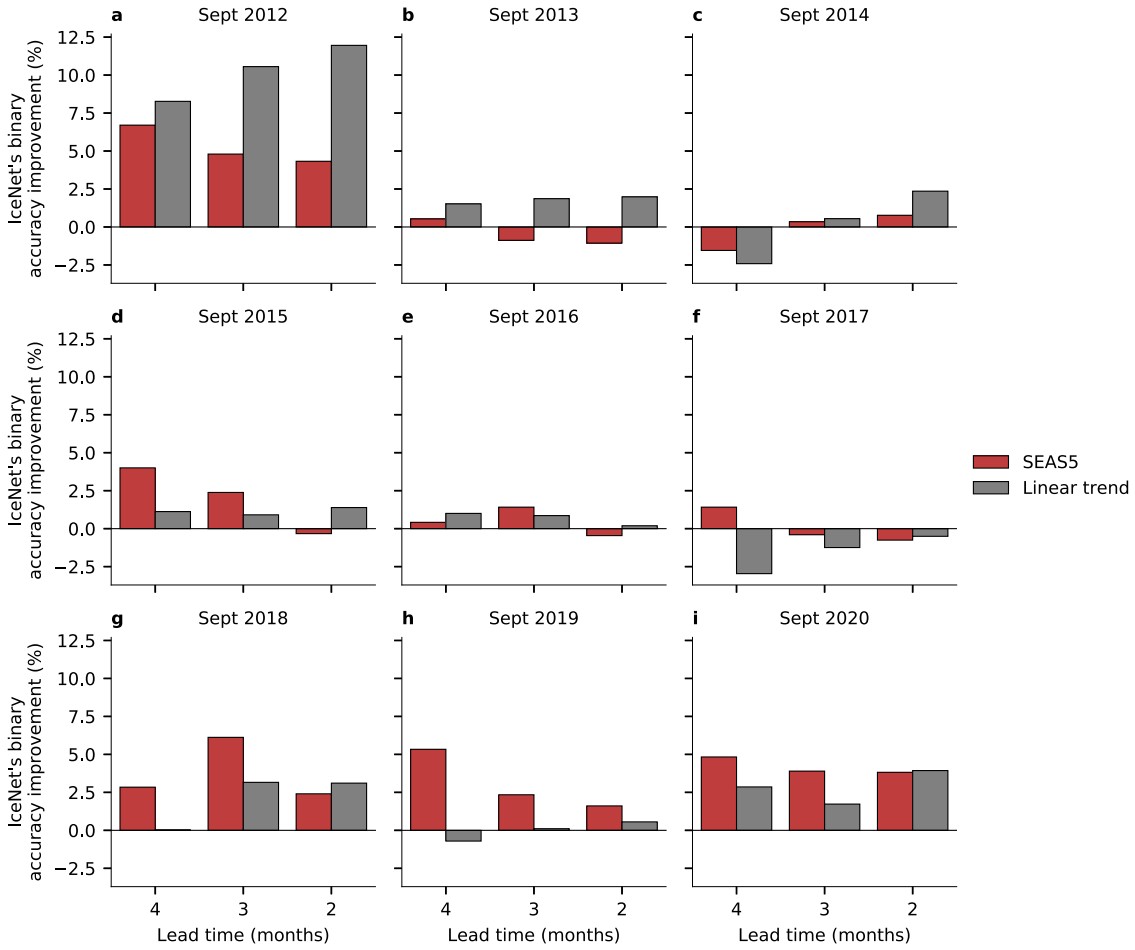

**Fig. 4 Comparing IceNet with SEAS5 and the linear trend for seasonal September forecasts. a–i** IceNet's improvement in binary accuracy relative to SEAS5 and the linear trend models for September forecasts at 4- to 2-month lead times for the validation and test years (2012–2020).

performs as well as or better than the other two models for this period (Fig. 3c, d).

A year-wise breakdown of IceNet's seasonal September binary accuracy relative to SEAS5 and the linear trend model is shown in Fig. 4. In the majority of the years 2012–2020, IceNet outperforms SEAS5, generally by a substantial margin. Where SEAS5 outperforms IceNet, it does so only by a small margin, underscoring the reliability and robustness of IceNet's September forecasts.

Figure 4 also provides information on IceNet's ability to forecast extreme changes in Arctic sea ice. A common metric used in sea ice analysis is sea ice extent (SIE), defined as the total area covered by grid cells with SIC > 15%. The 2012–2020 period contains three anomalous September SIEs: 2012 (lowest extent on record), 2013 (anomalously high extent), and 2020 (second-lowest extent on record). IceNet far outperforms the binary accuracy of SEAS5 and the linear trend in forecasting the extreme minimum extent years. IceNet's relative performance remains satisfactory for the high extent year of 2013 (Figs. 2e–g and 4b), despite a positive sea ice bias in SEAS5 and the linear trend that favours them in such years. These results indicate IceNet has particularly strong predictive capacity for extreme events relative to other models. Maps of IceNet's ice edge predictions for 2012, 2013, and 2020 are shown in Fig. 2.

The Sea Ice Outlook[49] (SIO) programme invites predictions for September SIE each year at 4-, 3-, and 2-month lead times. Comparing IceNet with the multi-model median September SIE predictions from the SIO shows that, on average, IceNet either matches or outperforms the SIO in terms of mean absolute SIE error over 2012–2020 (Supplementary Fig. 2a). A year-wise decomposition further reveals IceNet's good predictive skill for anomalous September ice extents when the SIO makes its largest errors[50] (2012, 2013, and 2020) (Supplementary Fig. 2b–j). These results provide a useful indicator of IceNet's performance relative to other models not included in this study. However, the absolute SIE error is a limited metric for forecast performance; it is the difference between the overpredicted area and the underpredicted area, and is thus a lower bound on the total misclassified area[47], which our binary accuracy metric measures. Therefore, the binary accuracy results in Fig. 4 provide a more robust assessment of IceNet's relative seasonal forecast skill for September.

**Effect of pre-training and ensembling**. The CMIP6 pre-training phase improves binary accuracy over the held-out years (2012–2020) by an average of 0.26%. This boost is small considering the increase in memory and computational load of the pre-training phase. The improvement is not uniform over each calendar month and lead time, even slightly hindering September forecasts for lead times longer than 3 months (Fig. 5a), potentially due to poor representations of summer melt processes of sea ice in the climate simulations. The mixed positive and negative effect of CMIP6 pre-training highlights the need for accurate process-based numerical models: improvements in physics-based modelling could translate into improvements in data-driven modelling as well. Unlike CMIP6 pre-training, model ensembling has a consistently positive effect, particularly for long-range summer

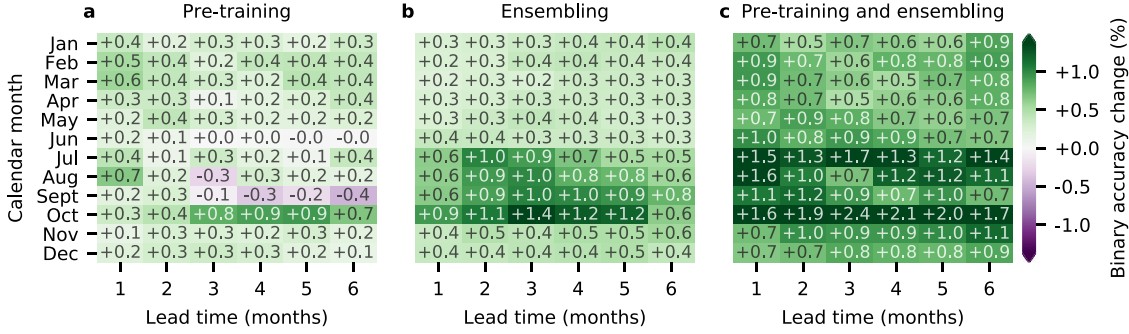

**Fig. 5 Quantifying the benefit of CMIP6 pre-training and ensembling in IceNet. a** IceNet's ensemble-mean binary accuracy relative to that of another 25-member ensemble without CMIP6 pre-training (i.e., training only on observational data). **b** IceNet's ensemble-mean binary accuracy relative to the mean binary accuracy of the 25 individual CMIP6 pre-trained ensemble members. **c** IceNet's ensemble-mean binary accuracy relative to the mean binary accuracy of the 25 individual ensemble members without CMIP6 pre-training. Each value is averaged over the validation and test years, 2012–2020.

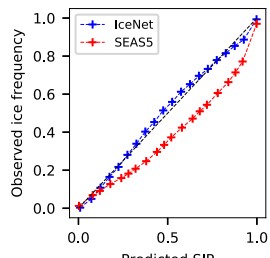

**Fig. 6 Calibration curves for IceNet and SEAS5.** Observed frequencies of ice (sea ice concentration >15%) are plotted against binned probabilities of the sea ice probability (SIP). IceNet's calibration curve is close to the ideal black dashed line, indicating that IceNet's output probabilities are almost equal to the true likelihood of sea ice occurring. In contrast, SEAS5 overestimates sea ice probability. The calibration curves are computed over the test years 2018–2020 and all six lead times.

predictions (Fig. 5b). The combined effect of CMIP6 pre-training and ensembling provides a significant boost to predictive skill (Fig. 5c), leading to IceNet's high performance.

**Probability calibration analysis**. Calibrated probabilities are highly desirable in probabilistic forecasting systems, with perfect calibration indicating the predicted probability equals the true probability of an event occurring. IceNet's SIP is almost perfectly calibrated over the test years (Fig. 6) as a result of ensembling and temperature scaling. In contrast, SEAS5—whose SIP is computed as the fraction of its 25 ensemble members with SIC > 15%— overpredicts the observed frequency of sea ice at all SIP values.

Calibration issues in dynamical models can be improved with a-posteriori calibration methods[22,37]: we used a simple bias correction scheme for SEAS5 due to ease of implementation (see Methods), although a more sophisticated method could bring further improvements to its accuracy and calibration.

**Bounding the ice edge**. Like calibration, sharpness is another useful diagnostic quality, referring to the degree to which a model's probabilities cluster around 0 or 1. Improving predictive performance can be framed as maximising sharpness subject to good calibration[51]. A well-calibrated and sharp sea ice forecasting model also enables the ice edge to be bounded between two contours of SIP, $p'$ and $1 - p'$, with $p = p'$ defining the maximum predicted ice edge location and $p = 1 - p'$ defining the minimum (see Methods). The choice of $p'$ involves a trade-off between reliability and precision (or spatial tightness) of the bounding region (Supplementary Fig. 3d). For IceNet, $p'_{90\%} = 0.036$ is a

reasonable choice, bounding 90% of the ice edge and 24.4% of the entire study area across all validation years and lead times. We use $p \in [p'_{90\%}, 1 - p'_{90\%}]$ to define an ice edge region in IceNet's forecasts. IceNet's binary accuracy is over 99% outside of the ice edge region, so we label $p \in [0, p'_{90\%})$ the confident open-water region and $p \in (1 - p'_{90\%}, 1]$ the confident ice region. This defines a new segmentation with the three aforementioned classes.

These findings are illustrated in Fig. 7, which shows IceNet's SIP forecasts and ice edge regions for the months of July, August and September 2020 at a 1-month lead time. Despite substantial changes in the spatial distribution of sea ice between months, IceNet's predicted ice edge is close to the observed ice edge (Fig. 7a–c), and its ice edge region reliably encompasses the observed ice edge (Fig. 7d–f).

A determining factor for ice edge bounding ability is the calibration of model probabilities close to $p = 0$ or $p = 1$. SEAS5 cannot bound the ice edge because it makes many errors at $p = 1$ (see Methods, Supplementary Figs. 3 and 4). Once this property is satisfied, the spatial precision of the ice edge region is determined by forecast sharpness. Analysis on the test years shows that the reliability of IceNet's ice edge region is stable with lead time due to an inflation of the area it covers (Supplementary Fig. 5). The inflation is greatest for forecasts that pass through the spring predictability barrier, corresponding to more uncertainty in the ice edge position (Supplementary Fig. 6). This suggests that limits on predictability lead to limits on the precision, and therefore usefulness, of probabilistic ice edge bounds. In contrast, applying this method to a daily sea ice forecasting system could provide tight bounds on the evolution of the ice edge position over weekly timescales when predictability is high. The framework developed here could play a role in ensuring safe shipping operations in the Arctic—which is expected to increase in coming decades—by helping ships avoid ice-covered waters[52].

**Variable importance analysis**. One key question is: 'How is IceNet using its input data to make predictions?'. We go some way to answer this by using a permute-and-predict method[53,54], which assigns an importance value to each input variable for each forecast, corresponding to the mean three-class accuracy drop when that variable is permuted (see Supplementary Methods). The top-5 most important input variables arising from this procedure for forecasting September and March at 5-, 3-, and 1-month lead times are shown in Table 1. The set of inputs IceNet is most sensitive to varies with lead time, depending more strongly on the linear trend forecast inputs at longer lead times. IceNet makes greater use of initial conditions as the initialisation month approaches the target month, especially for September forecasts. At a 1-month lead time, permuting the initialisation

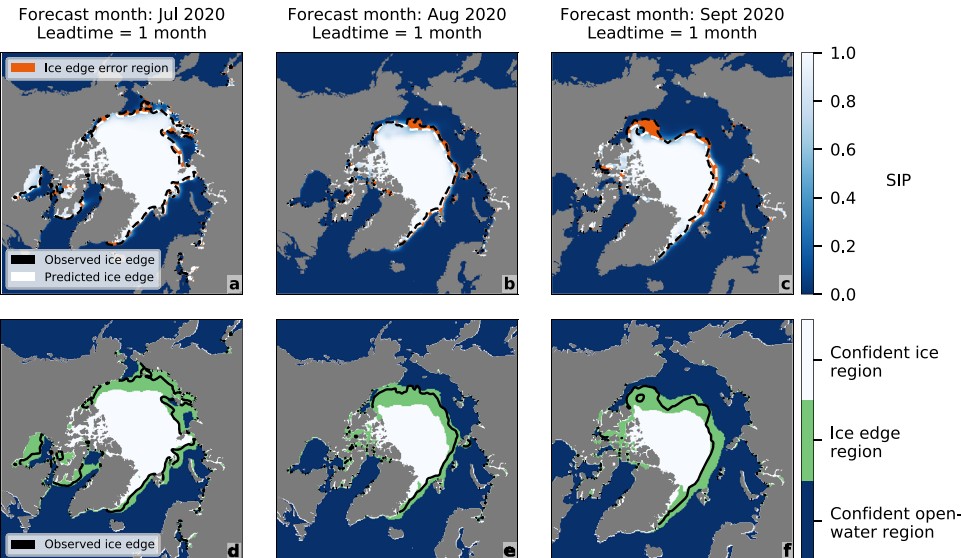

**Fig. 7 IceNet's forecasts for July, August, and September 2020 at a 1-month lead time. a–c** IceNet's predicted sea ice probability (SIP), $p$, with binary ice edge errors overlaid. **d–f** IceNet's predicted ice edge region, corresponding to $p \in [p'_{90\%}, 1 - p'_{90\%}]$.

SIC field results in the greatest accuracy drops for both March and September forecasts. The 1-month forecasts for September also depend on other synoptic conditions, such as the sea level pressure and 500 hPa geopotential height anomalies, which relate to tropospheric circulation patterns.

Deep learning systems like IceNet are adept at learning nonlinear statistical relationships between input and output data, but do not directly model causal relationships. Despite this, IceNet's permute-and-predict variable importance results are consistent with known causal links between climate variables and sea ice, suggesting that physically plausible statistical relationships have been learned. Furthermore, the diminishing importance of IceNet's initial conditions (relative to its linear trend forecast inputs) at greater lead times aligns with observed limits of sea ice predictability. This logic can be reversed: the predictability that IceNet can learn from training data provides evidence for the inherent timescales of memory in the Arctic climate system. For example, initial conditions are assigned negligible importance for IceNet's March forecasts initialised in October (5 month lead time) (Table 1b), with IceNet only improving upon the linear trend binary accuracy by an average of 0.3% (Fig. 3d). This suggests that memory in the Arctic system from October (early in freezing season) may be almost undetectable in the sea ice state by March (end of freezing season)—at least in the data that IceNet is presented with (Supplementary Table 2). In contrast, IceNet makes notable use of June initial conditions for its 3-months-ahead September forecasts (Table 1a), where it outperforms the linear trend binary accuracy by 2.1% (Fig. 3d). This strongly suggests that IceNet has learned how conditions in the middle of the melt-season can affect sea ice at the end of the melt season, evidenced further by IceNet's skill in forecasting anomalously high or low September ice extents at a 3-month lead time (Fig. 4 and Supplementary Fig. 2).

The full permute-and-predict results for each lead time, averaged over all calendar months, are reported in Supplementary Fig. 7. This reveals new patterns that are not apparent from the top-5 rankings in Table 1. For example, IceNet is sensitive to permutations of the initialisation 500 hPa geopotential height anomaly field, but not that of the 250 hPa geopotential height anomaly field, despite high correlation between the two variables. This could be because monthly averaged winds from the middle of the troposphere have a more dominant effect on sea ice than

monthly averaged winds from the top of the troposphere, and IceNet has learned to mostly ignore the 250 hPa input.

## Discussion

We have introduced an Arctic sea ice forecasting AI system, IceNet, which outperforms the leading dynamical model, SEAS5, in seasonal predictions of Arctic sea ice. A further benefit of IceNet is speed: once trained, IceNet runs over 2000 times faster on a laptop than SEAS5 running on a supercomputer, taking less than ten seconds on a single graphics processing unit. A variable importance method provided insight into the input variables IceNet uses to achieve state-of-the-art performance, with sea ice and tropospheric initial conditions being key in short-range predictions for September. IceNet's well-calibrated probabilistic forecasts enable the observed ice edge to be reliably bounded between two spatial contours of predicted sea ice probability, which could be used operationally to avoid shipping disasters, saving lives and preventing ocean contamination[52]. In addition, as the mechanisms between sea ice extent and Northern Hemisphere weather are better understood, accurate seasonal predictions of sea ice could anticipate weather conditions in the mid-latitudes months in advance.

A further significance of this work lies in posing a challenge to dynamical models. In regions and seasons where IceNet's forecasts outperform dynamical models, deficiencies in model fidelity or forecast initialisation are likely to be substantial. This information, combined with insight into which observations are the most important for IceNet's forecasts, provides valuable guidance for improving dynamical model parameterisations, data assimilation methods, and forecast calibration techniques.

While the implications of accurate sea ice forecasts for shipping are well developed[52], we argue that they could also play a pivotal role in adaptation and mitigation strategies for sea ice loss. Predictions for the timing and location of sea ice loss can provide early warnings for the possible sea ice conditions that lie ahead, which is critical for local communities, authorities, and Arctic ecosystem conservation groups. One example use case is with 'mega haul-outs' of Pacific walrus, occurring when tens of thousands of walrus are forced to congregate on land due to a lack of sea ice to rest on. Human disturbances can cause stampedes at haul-out sites and lead to high walrus mortality[55,56]. Information from IceNet's forecasts, combined with known haul-out locations,

**Table 1 Top-5 variable importance rankings from the permute-and-predict method.**

**(a)**

| Rank | Apr initialisation (5-month lead time) | | Jun initialisation (3-month lead time) | | Aug initialisation (1-month lead time) | |
|---|---|---|---|---|---|---|
| 1 | Sept linear trend SIC forecast | (−0.56%) | Jun upw. solar radiation anomaly | (−1.32%) | Aug SIC | (−22.94%) |
| 2 | Apr upw. solar radiation anomaly | (−0.51%) | Sept linear trend SIC forecast | (−0.74%) | Aug sea level pressure anomaly | (−2.55%) |
| 3 | Apr SIC | (−0.46%) | Jun SIC | (−0.63%) | Aug 500 hPa gpt. height anomaly | (−1.58%) |
| 4 | Apr 2 m air temperature anomaly | (−0.22%) | Aug linear trend SIC forecast | (−0.45%) | Aug upw. solar radiation anomaly | (−0.66%) |
| 5 | Feb 500 hPa air temperature anomaly | (−0.22%) | May 2 m air temperature anomaly | (−0.39%) | Jun 2 m air temperature anomaly | (−0.29%) |

**(b)**

| Rank | Oct initialisation (5-month lead time) | | Dec initialisation (3-month lead time) | | Feb initialisation (1-month lead time) | |
|---|---|---|---|---|---|---|
| 1 | Mar linear trend SIC forecast | (−0.28%) | Mar linear trend SIC forecast | (−0.50%) | Feb SIC | (−8.69%) |
| 2 | Apr linear trend SIC forecast | (−0.18%) | Feb linear trend SIC forecast | (−0.34%) | Feb 500 hPa gpt. height anomaly | (−0.58%) |
| 3 | Feb linear trend SIC forecast | (−0.14%) | Dec sea level pressure anomaly | (−0.30%) | Feb upw. solar radiation anomaly | (−0.40%) |
| 4 | Apr SIC | (−0.12%) | Apr linear trend SIC forecast | (−0.22%) | Apr linear trend SIC forecast | (−0.24%) |
| 5 | Oct sea level pressure anomaly | (−0.08%) | Apr SIC | (−0.09%) | Feb sea level pressure anomaly | (−0.21%) |

(a) For September forecasts with April, June, and August initialisations. (b) For March forecasts with October, December, and February initialisations. The mean three-class accuracy drops associated with each variable are shown in brackets. The input variables cosine of initialisation month index and sine of initialisation month index (Supplementary Table 2) were removed from the rankings as they do not have a physical interpretation (see Supplementary Methods). Gpt. = geopotential, upw. = upwelling.

could anticipate these events and help to prevent stampedes by managing human access. Furthermore, predictions for the migration of cetacean populations (which can coincide with sea ice advance and retreat[57]) could help to avoid fatal collisions between ships and endangered whale species. Such applications would help to fill an urgent gap in the integration of climate change in ecosystem management and planning tools[58]. The impacts of climate change on polar marine species and ecosystems, including the rapidly changing annual cycle of crucial sea ice habitat, means that dynamic approaches to conservation and management are imperative. For example, dynamic Arctic marine protected areas (MPAs) are likely to be more effective than those of static design[58]. Sea ice forecasts could inform the definition of such dynamic MPAs and provide advanced warning for stakeholders, allowing time to adapt activities to avoid areas critical for Arctic biodiversity. In such use cases, reliable quantification of uncertainty, as in IceNet forecasts, is likely to be crucial for the decision-making process.

IceNet demonstrates the potential of AI methods as a powerful tool for seasonal sea ice forecasting and an enabler of conservation planning tools in the Arctic. Future work will explore whether including ice thickness in IceNet's inputs improves its accuracy in summer. We will also implement a new online version of IceNet that operates on a daily temporal resolution, which is likely to improve performance at short lead times.

## Methods

**Training data considerations**. The datasets used for training IceNet comprise observational sea ice concentration (SIC), observational reanalysis data, and climate simulation data.

Satellites have measured sea ice conditions since late 1978 using passive microwave sensors. A number of different sensors have been used during this observation period, including the Scanning Multichannel Microwave Radiometer (SMMR) on NASA's Nimbus 7 satellite, the Special Sensor Microwave/Imager (SSM/I) on the Defence Meteorological Satellite Program's (DMSP) satellites, and the Special Sensor Microwave Imager and Sounder (SSMIS) on the later DMSP satellites. SIC can be computed from passive microwave satellite measurements using several different retrieval algorithms, between which substantial differences in the estimated SIC can arise. We obtained SIC data from the European Organisation for the Exploitation of Meteorological Satellites' (EUMETSAT) Ocean and Sea Ice Satellite Application Facilities (OSI-SAF) data record[59], comprising retrieval algorithms OSI-450 (1979–2015)[60] and OSI-430-b (2016 onwards)[61], which use data from SMMR (1978–1987), SSM/I (1987–2009) and SSMIS (2003–today) sensors. The OSI-450/OSI-430-b algorithms have been shown to be more accurate than other retrieval algorithms when compared with direct optical satellite observations of summer SIC (while the Arctic Ocean is sunlit)[34]. Owing to limitations in passive microwave measurements, no retrieval algorithm matches the true SIC. In particular, notable issues arise on the coastline due to land-sea spillover effects caused by snow on the land surface, which has a similar passive microwave signature to sea ice[62].

The OSI-450/OSI-430-b SIC dataset is provided on a Lambert Azimuthal Equal Area projection, with a grid spacing of 25 km. Also known as the Equal Area Scalable Earth 2 (EASE2) grid, this ensures areas on the Earth are preserved in the projection. The size of the SIC data is $432 \times 432$ on the EASE2 grid with each grid cell covering an area of 625 km$^2$. All other datasets considered in this study were re-gridded from a latitude-longitude grid to the EASE2 grid using bilinear interpolation.

For portions of the SIC data record, data surrounding the North Pole is missing due to satellite orbit and field of view restrictions. Known as the polar hole, the size of this data gap reduced over time as satellites were able to make observations closer to the North Pole. The missing area includes data north of 84° for SMMR data (1979 onwards), 87° for SSM/I data (1987 onwards), and 89° for SSMIS data (2003 onwards). We use bilinear interpolation to fill the SIC values in this region. These interpolated values form part of IceNet's input data, but we chose to not include them as training samples at the output of the networks.

Another source of gaps in the SIC dataset are missing daily observations due to satellite malfunctions, resulting in several months for which no monthly mean could be obtained: April–June 1986 and December 1987. Forecasts during the training years that depend on one of these missing months of SIC could not be made and thus were discarded from the training dataset. However, training forecasts that only included missing data at the output could be salvaged by masking out grid cells associated with the missing month from the training samples.

The non-SIC observational climate variables used as input to IceNet (Supplementary Table 2) are reanalyses obtained from ECMWF ERA5[63] at a 0.25° resolution. Reanalysis variables are based on data assimilation, combining observations with dynamical model data to form a consistent, gridded dataset using the laws of physics. We use the ERA5 monthly averaged data on single levels from 1979 to present dataset for surface variables[64], and the ERA5 monthly averaged data on pressure levels from 1979 to present dataset for the upper air variables[65].

The CMIP6 pre-training data was obtained from the Earth System Grid Federation (ESGF). Five simulations from the MRI-ESM2.0[66,67] ensemble were used: r1i1p1f1, r2i1p1f1, r3i1p1f1, r4i1p1f1, and r5i1p1f1. We also included five simulations from the EC-Earth3[68,69] ensemble: r2i1p1f1, r7i1p1f1, r10i1p1f1, r12i1p1f1 and r14i1p1f1. For each climate simulation, data from the historical and SSP2-4.5 experiments were concatenated to create a continuous time series for 1850–2100. The MRI-ESM2.0 and EC-Earth3 models were chosen because they included all the climate variables used for IceNet at a satisfactory resolution.

**Data preprocessing**. IceNet's non-SIC input variables with strong seasonal cycles, such as temperature and solar radiation, are converted to anomalies in order to emphasise differences from typical values. The anomaly variables were found by subtracting the climatological mean for each calendar month, computed over the observational training data period (1979–2011).

The speed and stability of neural network training can be improved by normalising the input data so that each variable takes values in similar ranges. We preprocessed the observational reanalysis variables by subtracting the mean and normalising by the standard deviation computed over the training years. To maintain direct correspondence between CMIP6 and ERA5 data, CMIP6 variables were normalised by the same mean and standard deviation values obtained from the ERA5 observational variables. The SIC data were converted from percentages in [0, 100] to fractions in [0, 1].

The full set of input-output samples, including the CMIP6 pre-training data, take up multiple terabytes in memory, which is too large to fit into RAM. To circumvent this issue, we built a custom data loader in Python to load batches of data on the fly while IceNet is trained—a standard approach for large training datasets in computer vision.

**SEAS5 ensembling and bias correction**. SEAS5's ensemble is generated by running multiple forecasts, each with small perturbations to the initial state and the model's internal parameters. Despite approximation of forecast uncertainty through an ensemble, fundamental limitations in a model's representation of physics lead to systematic forecasts errors known as 'bias', which calibration methods attempt to alleviate. We bias correct its ensemble-mean forecasts for 2012 onwards by subtracting the mean error field for a given calendar month and lead time, computed retrospectively by averaging over the years 2002–11.

**Description of the U-Net architecture adopted for IceNet**. IceNet is an ensemble of 25 CNNs[26]. The CNN architecture adopted for each IceNet ensemble member is a U-Net[29]. A U-Net is an encoder-decoder CNN where the feature-extracting encoding path of the network downsamples the input data, followed by a decoding path that upsamples the data. In IceNet, the output of the decoding path is fed into six different convolutional layers with linear activation functions and three feature maps each, corresponding to the 6 forecast months and three SIC classes. These feature maps are then divided by a temperature scaling parameter followed by a softmax activation function, mapping real values to probabilities that sum to 1 across the three SIC classes. In total, the IceNet architecture contains roughly 44 million trainable weights. IceNet's architecture is detailed in Supplementary Table 1 and illustrated in Fig. 1. We use batch normalisation in IceNet to speed up training and provide a small regularisation effect to reduce overfitting[70].

**Training procedure details**. Training IceNet begins with randomly initialising the network weights. We use He initialisation[71], which draws weights from a truncated Normal distribution with standard deviation dependent on the size of the previous layer. This helps in attaining a global minimum of the objective function faster and more efficiently. A different random seed was used for initialising each ensemble member, which results in different learned input-output mappings[35,72]. We used a focal loss[73] as the objective function for training, which is an extension of the common cross-entropy function for imbalanced classification problems. During training, randomly selected batches of training data are fed as input to the network, with the network targets defined as the sea ice concentration classes over the future 6 months (open-water: SIC ≤ 15%, marginal ice: 15% < SIC < 80%, full ice: SIC ≥ 80%). IceNet's weights were trained using backpropagation (gradient descent) of the focal loss with the Adam optimiser[74]. A batch size of 2 was used with an initial learning rate of 0.0005.

In winter months, 18% of the 432 × 432 grid cells on the EASE2 grid have a non-zero chance of sea ice occurring due to many grid cells being over land or too far south. In September, this drops to just over 10%. To avoid the loss function being dominated by trivial 0% SIC grid cells, we define an active grid cell region which shrinks in the summer and expands in the winter based on the maximum sea ice extent observed in a given calendar month. Samples outside of the active grid cell region are weighted by zero in the loss function. The active grid cell region was also used when computing accuracy metrics.

With no further modification to the loss function, the imbalance of samples towards the winter months would place more weight on forecasting winter during training. To ensure that each month contributes equally to the loss function, we use a month-wise weighting scheme based on the ratio of active grid cells relative to that of March. This results in September samples being weighted by a factor of 1.75 in the loss function, with the weighting decreasing to 1 for March.

Before training on observational data, we first use transfer learning by pre-training each IceNet ensemble member on the CMIP6 data. The entire CMIP6 dataset is presented to IceNet in a shuffled manner; this avoids fitting to one specific model's physics or to one time period at a given stage of training. After every 1000 pre-training batches, we compute the mean three-class accuracy of the network's forecasts over the observational validation years 2012–2017 and all six lead times. When the mean validation accuracy exceeds its previous best, the model's weights are checkpointed. Validating on the observational data during pre-training avoids overfitting to the CMIP6 models' representations of physics, each with their own systematic biases and limitations. Pre-training is run for two full passes through the pre-training dataset (i.e., two epochs).

After pre-training, the IceNet ensemble members are fine-tuned on the observational data training years, 1979–2011. We reduced the learning rate by a factor of 2 before fine-tuning and used a learning rate schedule that further reduced the learning rate by a factor of $e^{-0.1}$ per epoch after the first 3 epochs. To avoid overfitting to the observational data, we used the aforementioned mean validation accuracy to perform model checkpointing and early stopping with a patience of 10 epochs. This ensures the network's weights that give the best generalisation performance are used for the final model, before held-out performance starts to degrade due to overfitting.

IceNet was implemented in Python 3.7 using the deep learning library TensorFlow. All the computations were carried out using an Nvidia Quadro P4000 graphical processing unit (GPU). On our GPU, pre-training one ensemble model takes around one day, after which fine-tuning to the observational data can be done in 2 h or less.

**Hyperparameter tuning**. Hyperparameters like the learning rate, number of convolutional filters, and batch size all influence the training process or model. These can have a substantial impact on the performance of the trained model but cannot be learned by the training algorithm. To determine appropriate values for the initial learning rate, number of filters in each convolutional layer and the batch size, we employed an automated Bayesian hyperparameter tuning process[75] using the Python package wandb[76] (Weights and Biases), optimising for the mean validation accuracy.

**Ensembling approach**. IceNet is an ensemble of probabilistic predictors, as this has been shown to be a successful strategy for uncertainty quantification (in fact, outperforming Bayesian neural networks in terms of uncertainty quantification and out-of-distribution robustness)[35,72,77]. Specifically, we compute an ensemble-mean forecast by averaging the output probability distributions of each of the 25 ensemble members.

**Temperature scaling**. Modern neural networks are prone to miscalibration and are often systematically overconfident or underconfident[41]. To improve the calibration of IceNet's probabilistic forecasts we use temperature scaling[41], a simple post-hoc probability calibration method. Temperature scaling involves the inclusion of a single scalar parameter in the network, $T$, which divides the logits that are passed into the softmax activation function. Temperature scaling raises or lowers the entropy of the output probability distributions, making it systematically 'more uncertain' or 'less uncertain'—it has no effect on a model's most probable class and thus does not affect IceNet's three-class forecast accuracy. While temperature scaling can shift the SIP above or below 0.5 and thus affect binary accuracy and predicted SIE, we found it to have a negligible overall effect on those metrics over the validation years.

$T$ is fixed to a value of 1 during the training and is optimised using the finished model. We find $T$ in a scalar optimisation scheme using the Brent-Dekker method[78], with the objective function being the categorical cross-entropy over the validation years. We employ a two-stage temperature scaling process: firstly, we calibrate each trained IceNet ensemble member individually by optimising $\{T^{(i)}\}_{i=1}^{25}$ (using the same value of $T$ for each lead time); secondly, we calibrate the ensemble-mean model by optimising $\{T_e^{(k)}\}_{k=1}^{6}$ (using a different $T_e^{(k)}$ for each lead time $k$ in 1, 2, …, 6).

**Relationship between the binary accuracy metric and the integrated ice edge error**. The integrated ice edge error (IIEE), defined as the total area covered by grid cells where a binary error in predicting SIC > 15% was made, measures the closeness of the predicted and true ice edges[47]. Our binary

accuracy metric can be seen as a normalised version of the IIEE based on the following relationship:

$$\text{binary accuracy} = (1 - \text{IIEE}/\text{area of active grid cell region}) \times 100\%$$

Based on the areas covered by the active grid cell regions in September and March, a binary accuracy decrease of 1% corresponds to an IIEE increase of 120,000 km$^2$ in September and 210,000 km$^2$ in March.

**Bounding the ice edge**. Here we introduce the processing pipeline and statistical framework developed to bound the ice edge and relate this to forecast calibration and sharpness. Our approach relates to prior works that bound the classification frontier and link it to prediction uncertainty[79,80].

The pipeline begins by computing the ice edge contour positions for each validation and test month by finding the observed SIC = 15% contour and removing the contour segments along the coastline. Let $\lambda$ denote a binary variable indicating the ice edge position with $\lambda = 1$ at an ice edge contour grid cell and $\lambda = 0$ otherwise. The percentage of ice edge bounded by $[p', 1 - p']$ can be computed from the integral $\int_{p'}^{1-p'} \hat{P}(p \mid \lambda = 1)\mathrm{d}p$, where $\hat{P}(p \mid \lambda = 1)$ denotes the empirical estimate of the true probability distribution $P(p \mid \lambda = 1)$, which is the probability of obtaining a SIP output of $p$ from IceNet at a random grid cell, forecast month, and lead time where $\lambda = 1$. In practice, this corresponds to finding the percentage of $\lambda = 1$ cells bounded by $[p', 1 - p']$ (Supplementary Fig. 3a). Contours of $p'$ and $1 - p'$ collapse onto the predicted ice edge as $p' \to 0.5$ and encompass the entire study region when $p' = 0$. Hence, increasing $p'$ decreases the fraction of all grid cells bounded by $[p', 1 - p']$: $\int_{p'}^{1-p'} \hat{P}(p)\mathrm{d}p$ (Supplementary Fig. 3b). This results in a trade-off between reliability of the ice edge region and the spatial tightness of the bound (Supplementary Fig. 3d). Analysis on the validation years shows that bounding 90% of the ice edge corresponds to $p'_{90\%} = 0.036$, which in turn corresponds to 24.4% of all grid cells. We label the grid cells with $p \in [p'_{90\%}, 1 - p'_{90\%}]$ as IceNet's ice edge region.

The ice edge bounding ability can be framed in terms of forecast calibration and sharpness by relating it to bounding the binary ice edge errors. The binary errors are themselves bounded by the predicted ice edge ($p = 0.5$) and the observed ice edge (Fig. 7a–c), leading to a nonlinear, monotonically increasing relationship between ice edge bounding and binary error bounding (Supplementary Fig. 3e). Let the binary variable $e$ indicate binary error locations, where $e = 1$ if a binary error occurred and $e = 0$ if the correct class was predicted. The percentage of binary errors bounded by $[p', 1 - p']$ is given by $\int_{p'}^{1-p'} \hat{P}(p \mid e = 1)\mathrm{d}p$. The distribution of SIP over error grid cells, $P(p \mid e = 1)$, relates to the forecast calibration and sharpness through Bayes' Rule: $P(p \mid e = 1) \propto P(e = 1 \mid p) \cdot P(p)$, where $P(e = 1 \mid p)$ measures calibration and $P(p)$ measures sharpness. Supplementary Fig. 4 plots empirical estimates of these distributions for IceNet and SEAS5 over the validation years. Ability to bound the binary forecast errors (and thus ability to bound the observed ice edge) is therefore governed by forecast calibration and sharpness. Particularly crucial is the calibration of model probabilities close to $p = 0$ or $p = 1$ because the majority of forecasts are made there (Supplementary Fig. 4b). If too many binary errors are made close to 0 and/or 1, the model will only be able to bound a sufficient fraction of the ice edge with a small $p'$ and thus a large, uninformative ice edge region. Owing to miscalibration, SEAS5 makes many errors at $p = 1$ (Supplementary Fig. 4f), preventing the model from bounding the ice edge (Supplementary Fig. 3a). The discretisation of SIP with dynamical models also hampers the fidelity with which a suitable ice edge region can be chosen, calling for a focus on continuous probability models like IceNet for ice edge bounding purposes.

An alternative scheme would be to train a binary classification model to predict the probability of a grid cell containing the ice edge, $P(\lambda = 1)$, and finding an appropriate threshold $P(\lambda = 1) \geq p'$ for the ice edge region, as above. However, the distinction between SIC ≤ 15% and SIC > 15% classes in IceNet has added utility over predicting the boundary between those classes alone.

## Data availability
The trained IceNet network weights, IceNet's ensemble-mean SIP forecasts, the forecast analysis results, the permute-and-predict results, and the ice edge bounding results have been deposited on the Polar Data Centre[81] (https://doi.org/10.5285/71820E7D-C628-4E32-969F-464B7EFB187C). The other datasets used in this paper comprise observational SIC data, observational reanalysis, climate simulations, and SEAS5 historical forecasts, all of which are available online. The observational SIC data is provided by OSI-SAF (http://osisaf.met.no/p/ice/). The reanalysis data was obtained from ERA5 (single level variables: https://cds.climate.copernicus.eu/cdsapp#!/dataset/reanalysis-era5-single-levels-monthly-means; pressure level variables: https://cds.climate.copernicus.eu/cdsapp#!/dataset/reanalysis-era5-pressure-levels-monthly-means). CMIP6 data is available via ESGF (https://esgf-node.llnl.gov/projects/esgf-llnl/). SEAS5 forecasts can be obtained at 0.25° resolution from the ECMWF MARS archive (https://www.ecmwf.int/en/forecasts/datasets).

## Code availability
IceNet was implemented using TensorFlow (https://www.tensorflow.org). The code to fully reproduce the paper's results are available at https://github.com/tom-andersson/icenet-paper[82] (https://doi.org/10.5281/zenodo.5176573). The code includes downloading and

preprocessing all the data used in the study; setting up the IceNet data loader for batch generation; IceNet's architecture and training; hyperparameter tuning; probability calibration; forecast accuracy analysis; the permute-and-predict algorithm; assessing ice edge bounding ability; downloading data generated from the study; and reproducing the paper's figures.

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

## Acknowledgements

This work is supported by Wave 1 of The UKRI Strategic Priorities Fund under the EPSRC Grant EP/T001569/1, particularly the 'AI for Science' theme within that grant and The Alan Turing Institute. D.J. is supported by a UKRI Future Leaders Fellowship (grant reference MR/T020822/1). D.J. and J.S.H. are supported by the NERC ACSIS project (grant NE/N018028/1). Y.A. is supported by the NERC National Capability programmes LTS-M ACSIS (North Atlantic climate system integrated study), grant NE/N018044/1, and LTS-S CLASS (Climate–Linked Atlantic Sector Science), grant number NE/R015953/1, by Advective Pathways of nutrients and key Ecological substances in the Arctic (APEAR) project (NE/R012865/1, NE/R012865/2, #03V01461), part of the Changing Arctic Ocean programme, jointly funded by the UKRI Natural Environment Research Council (NERC) and the German Federal Ministry of Education and Research (BMBF). Y.A. is also supported by funding from the European Union's Horizon 2020 research and innovation programme under project COMFORT (grant agreement no. 820989), for which the work reflects only the authors' view; the European Commission and their executive agency are not responsible for any use that may be made of the information the work contains). Y.A. is also supported by the NERC projects "Towards a marginal Arctic sea ice cover" (grant NE/R000085/1) and "PRE-MELT" (Grant NE/T000546/1). E.B.W. is supported by NSF (OPP grant 1751363). We thank Fruzsina Agocs (Kavli Institute for Cosmology) for early discussions and advice on data loaders. We thank Bablu Sinha (National Oceanography Centre), Christine McKenna (University of Leeds), and Hua Lu (British Antarctic Survey) for advice on atmospheric climate variables. We also thank Larry Hamilton (University of New Hampshire) for providing the historical SIO forecast data. We also thank Rachel Cavanagh (British Antarctic Survey) for discussions on sea ice forecasts for dynamic marine protected areas. This work was generated using ERA5 data from the Copernicus Climate Change service. Neither the European Commission nor ECMWF is responsible for use of the Copernicus data.

## Author contributions

T.R.A. designed, built and trained IceNet with guidance from M.P.-O., B.P., A.E., C.R. and S.L. T.R.A. produced the figures and wrote the code. T.R.A. developed the ice edge bounding framework with contributions from A.E. T.P. assisted with ERA5, CMIP6 and sea ice data access and preprocessing. J.B. improved the efficiency of the training process and assisted with experiments for the paper's revision. D.J., Y.A. and J.W. advised on climate variables for IceNet's inputs. S.T. and B.S. provided guidance on the SEAS5 model and bias correction. E.B.-W. facilitated the Sea Ice Outlook comparison. R.D. conceptualised the impacts of IceNet for Arctic conservation work. T.R.A., J.S.H. and M.-P.O. wrote the paper with contributions and feedback from all authors. J.S.H. and E.S. wrote the funding proposal for the work. J.S.H. managed the project.

## Competing interests

The authors declare no competing interests.
