## [Peer Review File · Nature Communications]

REVIEWER COMMENTS

Reviewer #1 (Remarks to the Author):

I know little about ML so it is a little hard for me to fully understand all content and details of the study. Overall, I enjoyed reading this paper, and I think it would make a good contribution to our field. My concerns were mostly minor and centered on the selection of 11 input variables to train IceNet. I've suggested minor revisions which are detailed below.

It is not clear to me why those 11 variables were selected as the key input to train their model. It seems that some of these variables have very severe collinearity or multilinearity issues. For example, 200 hpa geopotential height, 500 hPa temperature and height, and SLP are variables with very high linear correlations on year to year time scales. SW down, SW up and sea ice condition (SIC) are also correlated to some extent. In addition, some key variables that be long known to have strong impacts on sea ice melting are not included in this table, such as specific humidity, clouds, downward long wave radiation, etc. I am not sure whether the authors have tried to extensively search all possible variables that could further add new value and skill on their model.

As far as I know, overfitting is a big concern for all ML studies. But authors seem to be worry-free in this regard. I am wondering why this concern doesn't need a particular attention here.

In the abstract (line 35), I don't understand what "two contours" refer to when I first read this part. Some observed sea ice records are available before 1979 and I think this extra data source could be used to validate IceNet. Currently only a few years of observations (2012-2020) are used to evaluate their model, which leads me to doubt a bit about the robustness of their conclusion and findings.

Line 33-34: "It also demonstrates a greater ability to predict anomalous pan-Arctic sea ice extents than the models submitted to the Sea Ice Outlook programme: I don't think this statement can hold well since it appears to me that the IceNet's performance is very close to the overall performance of all models in the SIO (Fig.3). The SIO is a little better than IceNet in 2 and 4-month leads, actually. The authors purposely compare IceNet with a single dynamical model (SEAS5). In my view, a more appropriate evaluation should be made to compare IceNet with all dynamical models participating in the SIO since it is unknown where SEAS5 lies in the spectrum of overall performance derived from all available dynamical models in the community.

Reviewer #2 (Remarks to the Author):

The authors present a deep learning sea ice forecasting system for seasonal predictions of probabilistic sea ice cover up to 6-months in advance. This system performs favorably when compared to a dynamical model in some instances and to many of the forecasts from the Sea Ice Outlook.

The paper is well written, and the work presents a novel approach to sea ice forecasting. However, there is some clarification that I feel should be made before I can recommend publication. My main issue is with the choice of predictors, colinearly among predictors, and the method of variable importance. While the choice of predictors may seem logical for their relation to changes in sea ice, some references here on why these variables were chosen should be included.

Table 1a and 1b show the drop in accuracy for the top 5 variables. The 5th ranked variables' drop in accuracy is roughly $\sim 0.2\%$. I am curious what lower ranked variables score? It seems many of the variables might be having quite a negligible (maybe even harmful) affect on the predictions. Did you use this method to prune variables that are not helpful? I would imagine this could improve computation costs or allow you to explore other variables that have proven to be influential on the evolution of sea ice (i.e., longwave radiation, specific humidity, vapor transport). As you mention, knowledge of which variables are influential for sea ice forecasting could be useful for improving

dynamic models, but knowledge of which variables have no influence could be of use as well.

How does this method handle collinearity among input variables? You mention that removing the seasonal cycle from non-SIC variables removes a lot of correlation, but I believe some variable anomalies would likely still be highly correlated (same variable at different pressure levels, air temperature and geopotential height...).

Line by line comments:

Line 77: Why start with three classes then reduce to two? Why not just find the probability of ice/no ice? I might be misunderstanding this process, so either way I think it should be explained more clearly.

Line 95: How sensitive are the results to the choice of emission scenarios?

Line 106: Use a linear trend of what? Sea ice concentration? How do you get a probabilistic forecast from this for comparison to your model's output?

Line 109: Is it not problematic to use the linear trend as a predictor and then compare your output to the linear trend? Wouldn't your model necessarily outperform the linear trend in this case?

Lines 342-343: Remove "The active grid cell area s."

Line 642: Did you look at the distribution of the accuracy drop values? Are they normally distributed? Are there outliers? Would the median work just as well as the mean?

Reviewer #3 (Remarks to the Author):

The authors proposed a new framework, IceNet, to forecast sea ice information. The technique employed and result validation are sound and thorough. It is a great pleasure to read this paper. Still, the reviewer may wonder if there are any further conclusions that can be drawn from this paper/presented results in the aspect of environment or climate changes? Or other insights beyond the technical highlights. In addition, the reviewer has the following minor comments for the authors to consider.

L65-66 The authors mention sea ice sensing using CNN. It may be more appropriate to include associated work as refs. such as

Q. Yan and W. Huang, "Sea ice sensing from GNSS-R data using convolutional neural networks," *IEEE Geosci. Remote Sens. Lett.*, vol. 15, no. 10, pp. 1510–1514, Oct. 2018.

and

L. Wang, K. A. Scott, L. Xu, and D. A. Clausi, "Sea ice concentration estimation during melt from dual-pol SAR scenes using deep convolutional neural networks: A case study," *IEEE Trans. Geosci. Remote Sens.*, vol. 54, no. 8, pp. 4524–4533, Aug. 2016.

L106-110 Is there any selection of forecast to be passed into the IceNet based on the performance of the corresponding linear trend model?

Reviewer comments

Editor comment

We agree with the reviewers that the collinearity and potential over-fitting issues related to the model inputs need to be more thoroughly addressed, supported by additional validation and sensitivity analysis. Likewise, please give attention to Reviewer #3's suggestion for an expanded discussion on the implications beyond the technical improvements of your new model.

When resubmitting, you must provide a point-by-point response to the reviewers' comments. Please show all changes in the manuscript text file with track changes or colour highlighting. If you are unable to address specific reviewer requests or find any points invalid, please explain why in the point-by-point response.

Review #1

I know little about ML so it is a little hard for me to fully understand all content and details of the study. Overall, I enjoyed reading this paper, and I think it would make a good contribution to our field. My concerns were mostly minor and centered on the selection of 11 input variables to train IceNet. I've suggested minor revisions which are detailed below.

It is not clear to me why those 11 variables were selected as the key input to train their model. It seems that some of these variables have very severe collinearity or multilinearity issues. For example, 200 hpa geopotential height, 500 hPa temperature and height, and SLP are variables with very high linear correlations on year to year time scales. SW down, SW up and sea ice condition (SIC) are also correlated to some extent. In addition, some key variables that be long known to have strong impacts on sea ice melting are not included in this table, such as specific humidity, clouds, downward long wave radiation, etc. I am not sure whether the authors have tried to extensively search all possible variables that could further add new value and skill on their model.

As far as I know, overfitting is a big concern for all ML studies. But authors seem to be worry-free in this regard. I am wondering why this concern doesn't need a particular attention here.

In the abstract (line 35), I don't understand what "two contours" refer to when I first read this part.

Some observed sea ice records are available before 1979 and I think this extra data source could be used to validate IceNet. Currently only a few years of observations (2012-2020) are used to evaluate their model, which leads me to doubt a bit about the robustness of their conclusion and findings.

Line 33-34: “It also demonstrates a greater ability to predict anomalous pan-Arctic sea ice extents than the models submitted to the Sea Ice Outlook programme: I don’t think this statement can hold well since it appears to me that the IceNet’s performance is very close to the overall performance of all models in the SIO (Fig.3). The SIO is a little better than IceNet in 2 and 4-month leads, actually.

The authors purposely compare IceNet with a single dynamical model (SEAS5). In my view, a more appropriate evaluation should be made to compare IceNet with all dynamical models participating in the SIO since it is unknown where SEAS5 lies in the spectrum of overall performance derived from all available dynamical models in the community.

Review #2

The authors present a deep learning sea ice forecasting system for seasonal predictions of probabilistic sea ice cover up to 6-months in advance. This system performs favorably when compared to a dynamical model in some instances and to many of the forecasts from the Sea Ice Outlook.

The paper is well written, and the work presents a novel approach to sea ice forecasting. However, there is some clarification that I feel should be made before I can recommend publication. My main issue is with the choice of predictors, colinearity among predictors, and the method of variable importance. While the choice of predictors may seem logical for their relation to changes in sea ice, some references here on why these variables were chosen should be included.

Table 1a and 1b show the drop in accuracy for the top 5 variables. The 5th ranked variables’ drop in accuracy is roughly ~0.2%. I am curious what lower ranked variables score? It seems many of the variables might be having quite a negligible (maybe even harmful) affect on the predictions. Did you use this method to prune variables that are not helpful? I would imagine this could improve computation costs or allow you to explore other variables that have

proven to be influential on the evolution of sea ice (i.e., longwave radiation, specific humidity, vapor transport). As you mention, knowledge of which variables are influential for sea ice forecasting could be useful for improving dynamic models, but knowledge of which variables have no influence could be of use as well.

How does this method handle collinearity among input variables? You mention that removing the seasonal cycle from non-SIC variables removes a lot of correlation, but I believe some variable anomalies would likely still be highly correlated (same variable at different pressure levels, air temperature and geopotential height...).

Line by line comments:

Line 77: Why start with three classes then reduce to two? Why not just find the probability of ice/no ice? I might be misunderstanding this process, so either way I think it should be explained more clearly.

Line 95: How sensitive are the results to the choice of emission scenarios?

Line 106: Use a linear trend of what? Sea ice concentration? How do you get a probabilistic forecast from this for comparison to your model's output?

Line 109: Is it not problematic to use the linear trend as a predictor and then compare your output to the linear trend? Wouldn't your model necessarily outperform the linear trend in this case?

Lines 342-343: Remove "The active grid cell area s."

Line 642: Did you look at the distribution of the accuracy drop values? Are they normally distributed? Are there outliers? Would the median work just as well as the mean?

Review #3

The authors proposed a new framework, IceNet, to forecast sea ice information. The technique employed and result validation are sound and thorough. It is a great pleasure to read this paper. Still, the reviewer may wonder is there any further conclusions can be drawn

from this paper/presented results in the aspect of environment or climate changes? Or other insights beyond the technical highlights. In addition, the reviewer has following minor comments for the authors to consider.

L65-66 The authors mention sea ice sensing using CNN. It may be more appropriate to include associated work as refs. such as

Q. Yan and W. Huang, "Sea ice sensing from GNSS-R data using convolutional neural networks," *IEEE Geosci. Remote Sens. Lett.*, vol. 15, no. 10, pp. 1510–1514, Oct. 2018.
and

L. Wang, K. A. Scott, L. Xu, and D. A. Clausi, "Sea ice concentration estimation during melt from dual-pol SAR scenes using deep convolutional neural networks: A case study," *IEEE Trans. Geosci. Remote Sens.*, vol. 54, no. 8, pp. 4524–4533, Aug. 2016.

L106-110 Is there any selection of forecast to be passed into the IceNet based on the performance of the corresponding linear trend model?

Responses to reviews

Note: Reviewers' comments appear in **black text**. Our replies appear in **blue text**. The manuscript has been revised using tracked changes, with **blue text** for additions and ~~strikethrough~~ for deletions.

Overview

We thank the reviewers for their insightful and helpful suggestions for our manuscript, and we appreciate the reviewers' unanimous recognition of the novelty of our work. The clarification requests and suggestions for minor revisions have helped us to improve both the clarity and robustness of our manuscript. Below this overview, we reply to each comment in a point-by-point fashion, endeavouring to be as thorough as possible and outlining any changes made to the manuscript.

It may be helpful to give a quick summary of the main components of our revision. We have added two new figures to the Supplementary Information, which provide additional validation of IceNet's performance and further insights into the input variables IceNet is most sensitive to. Furthermore, a new figure has been added to the main text to provide a robust breakdown of IceNet's seasonal September forecasting ability and ability to predict extreme events. This has replaced our figure based on the Sea Ice Outlook (SIO), which we have moved to the Supplementary Information. Several paragraphs have been included in the main text to discuss these additional results. The text has also been updated with a new section on our choice of input variables, which has improved the flow of our manuscript, answering questions that many readers will have. Finally, the authorship has conducted another round of scrutiny of the manuscript, and minor changes to the language have been made in some places to ensure maximum clarity of the text (for example, renaming 'active grid cell area' to 'active grid cell region'). As some figures have been rearranged, all references to figures reflect figure numbering in the revised manuscript.

We hope our responses and revisions satisfy the reviewers, and that our revised manuscript will be considered for publication in Nature Communications.

Reviewer 1 responses

Comment 1

It is not clear to me why those 11 variables were selected as the key input to train their model.

Response 1

We are grateful to the reviewer for pointing out that our reasoning behind the choice of 11 input variables for IceNet was not sufficiently explained in our initial submission, a comment also made by Reviewer 2 (R2). We have added a new section to the text in response to this concern and we refer the reviewer to Response 1 to R2 for further details.

Comment 2

It seems that some of these variables have very severe collinearity or multilinearity issues. For example, 250 hpa geopotential height, 500 hPa temperature and height, and SLP are variables with very high linear correlations on year to year time scales. SW down, SW up and sea ice condition (SIC) are also correlated to some extent.

Response 2

The reviewer is correct to observe that there are large correlations between several of our input variables - a concern also raised by R2. Multicollinearity can significantly affect the performance of simple linear models by inflating the variance of the model's linear coefficients, which can harm extrapolation behaviour. However, neural networks are insensitive to multicollinearity because they are significantly overparameterised, leading to redundancy in the individual weight values (De Veux & Ungar 1994). We are confident that IceNet is not sensitive to correlations in its input data because the neural network used for IceNet is very large (containing over 44 million weights). This is evidenced by our sensitivity experiment testing the inclusion of additional variables to IceNet (see Comment 3). In this experiment, including additional variables with multicollinearity (longwave surface radiation downwards and cloud cover) had no impact on forecasting skill.

Comment 3

In addition, some key variables that be long known to have strong impacts on sea ice melting are not included in this table, such as specific humidity, clouds, downward long wave radiation, etc. I am not sure whether the authors have tried to extensively search all possible variables that could further add new value and skill on their model.

Response 3

We thank the reviewer for raising interest in our selection of predictors for IceNet and for suggesting additional input variables, which was very similar to the list suggested by R2.

In response, we performed an ablation experiment (systematic removal of a component of an AI system) to test how IceNet's performance changes with the inclusion of ERA5 downward longwave surface radiation, cloud cover, and specific humidity. To account for the randomness of the training procedure, we trained two ensembles of 5 networks: a baseline ensemble with IceNet's standard input variables (Supplementary Table 1), and another ensemble with the three additional variables. To reduce computational costs, each network was trained on the observational data only (no CMIP6 pre-training). Our results of binary accuracy vs. lead time over the validation and test years (2012-2020) show that IceNet is insensitive to the inclusion of a few extra variables (Rebuttal Fig. 1). For comparison, we also plot the binary accuracy of IceNet and the linear trend model. The difference between the ensembles from this experiment and IceNet can be explained by the greater number of ensemble members (5 for this experiment, 25 for IceNet), and the use of CMIP6 pre-training in IceNet (Fig. 6).

In order to extensively search all possible input variables, as suggested by the reviewer, we would have to run a multi-pass drop-and-relearn experiment (Hooker & Mentch, 2019). In the context of IceNet, drop-and-relearn starts off by sequentially dropping one input variable at a time and measuring the accuracy on the validation data for a particular lead time. Due to the randomness of training, the training run associated with each variable drop must be performed multiple times with different random seeds, and the validation accuracy averaged across seeds. The variable drop corresponding to the greatest accuracy (i.e. lowest accuracy drop) is then removed from the model and assigned the lowest rank. The experiment then proceeds to the next pass, dropping one variable at a time and removing the least important, and so on until there are only two variables left (which determines the first and second place in the ranking). The resultant ranking provides a rigorous assessment of which input variables are the most important for the performance of the learning algorithm. Drop-and-relearn does not suffer from correlated variables being assigned inflated importance, as with permute-and-predict (Hooker & Mentch, 2019).

Rebuttal Fig. 1 | Mean binary accuracy versus lead time over the validation and test years (2012-20) shown for a 5-member baseline ensemble (without CMIP6 pre-training) and an identical 5-member ensemble but with three additional input variables. IceNet and the linear trend model are also shown for reference.

Unfortunately, multi-pass drop-and-relearn is extremely computationally expensive due to the volume of training runs required. As such, it would only be possible to train on the observational record (i.e., the CMIP6 pre-training step must be removed). We implemented the code for this experiment after first making our training runs as fast as possible, reaching ~1 hour per training run. With the inclusion of several additional variables as input to IceNet based on the reviewer’s suggestions (downward longwave radiation, cloud cover, and specific humidity), the total number of input variables is 19 (grouping across lags/leads), resulting in 190 variable drops per drop-and-relearn ranking. We would need to run the experiment six times to determine a different ranking for each lead time, because the variable importance depends greatly on lead time (as determined from our permute-and-predict results). The total time cost for this experiment (including HPC queueing and careful analysis of the results) could exceed 6 months, which is unfeasible given our resources. As such, we decided to postpone this experiment from the revision of this manuscript to a future study. We are grateful to R1 and R2 for inspiring a follow-on project.

Comment 4

As far as I know, overfitting is a big concern for all ML studies. But authors seem to be worry-free in this regard. I am wondering why this concern doesn't need a particular attention here.

Response 4

We agree with the reviewer that overfitting is one of the main concerns in classical machine learning. Overfitting happens when a model learns the detail and noise in the training data to the extent that it negatively impacts the performance of the model on new data. However, deep learning models have been proven to generalise well to unseen data while still fitting the training data exactly, unlike with classical methods. We refer the reviewer to Belkin 2019 where the following excerpt of the abstract is relevant to this discussion:

“The bias–variance trade-off implies that a model should balance underfitting and overfitting: Rich enough to express underlying structure in data and simple enough to avoid fitting spurious patterns. However, in modern practice, very rich models such as neural networks are trained to exactly fit (i.e., interpolate) the data. Classically, such models would be considered overfitted, and yet they often obtain high accuracy on test data.”

IceNet's high performance on the validation and test datasets, demonstrated thoroughly in this study, shows that our method generalises well.

Overfitting can be easily identified by checking validation metrics (e.g. mean binary accuracy). The validation metrics usually improve with continued training until a point where they stagnate. After enough training epochs the metrics may worsen: this is when overfitting starts to occur, which can happen to deep learning systems. To mitigate this, we employ the established technique of *early stopping* in our study. The early stopping procedure is mentioned on Line 432 in the Methods. In response to the reviewer's concern, we have added a sentence at Line 432 to highlight this procedure.

Comment 5

In the abstract (line 35), I don't understand what "two contours" refer to when I first read this part.

Response 5

We appreciate the feedback on the clarity of the text. To avoid confusion, this sentence has now been removed from the abstract in order to meet the 150-word limit of Nature Communications. When we refer to IceNet's ice edge bounding ability in the Discussion, we have made the following addition to the text on Line 286: '...enable the observed ice edge to be reliably bounded between two spatial contours **of predicted sea ice probability**'.

Comment 6

Some observed sea ice records are available before 1979 and I think this extra data source could be used to validate IceNet. Currently only a few years of observations (2012-2020) are used to evaluate their model, which leads me to doubt a bit about the robustness of their conclusion and findings.

Response 6

There are no continuous satellite concentration records before SSMR (the Scanning Multichannel Microwave Radiometer), which became operational in Oct 1978 (Cavalieri et al. 1996). The lack of satellite observations before this date significantly hinders the reliability of any gridded model reanalysis data for sea ice concentration prior to 1979. As the reviewer suggests, there are some spatially and temporally sparse observations before 1979, e.g. from submarine and airborne measurements (see Fig. 2 of Johnson et al. 2012). However, these fieldwork measurements are not pan-Arctic, typically only observing sea ice thickness along mostly linear tracks. Therefore, these datasets would not be appropriate sources either for running IceNet forecasts or for validating forecasting skill on 25x25 km gridded sea ice concentration data.

We agree with the reviewer that more years for model evaluation would support the evaluation of IceNet's performance. However, given the limited observational record, our choice of 9 years for evaluation out of the total of 42 years reflects a trade-off between having

enough training data to obtain a performant model and having enough validation data to be confident in its extrapolation ability. Dynamical sea ice models typically require some form of bias correction to account for severe limitations in model physics, which will be computed retrospectively over a hindcast period, and is analogous to our training dataset. Therefore, dynamical models will too require a partition of the observational record between calibration and evaluation periods.

Other sea ice forecasting model evaluation studies have used a similar number of years for evaluation (12 years, Zampieri et al. 2018), and even substantially less (11 months, Wayand et al. 2018). We also point out that our evaluation period of 2012-2020 comprises 21% of the entire dataset and 2.76 million active sea ice grid cells (see Lines 409-411 for a definition of the active grid cell region), and so each point on Fig. 3a is computed over a very large number of grid cells. Furthermore, as Arctic sea ice continues to thin and the dominant melt processes change, summer sea ice is becoming increasingly non-stationary and difficult to predict (Guemas et al. 2014), making the 2012-2020 period a particularly challenging setting in which to evaluate our new model. Therefore, we feel that extending the held-out evaluation dataset further into the past (i.e., before 2012) would not substantially aid model evaluation.

Our strongest claims in the manuscript are regarding IceNet's superior seasonal summer forecasting performance. As such, we have produced an alternative representation of Fig. 3 to provide further validation to our claims (Rebuttal Fig. 2). The figure shows the number of evaluation years where IceNet outperformed SEAS5 (left) and the linear trend (right) in terms of binary accuracy. For 3-month September forecasts, IceNet outperformed SEAS5 in 7/9 years and the linear trend in 8/9 years. Furthermore, IceNet outperformed both benchmarks in all 3 of the test years (2018-2020) for 2-5 month September forecasts (not shown). This highlights the consistency with which IceNet made the strongest seasonal summer forecasts of the two benchmarks, and reveals strong relative performance in seasonal winter forecasts as well. We believe this new analysis provides further evidence for the robustness of our claims. As such, we have included it in the manuscript as Supplementary Figure 5. We refer to the figure in Line 159 in the 'Evaluation of IceNet's performance' section.

Rebuttal Fig. 2 | The number of years that IceNet’s binary accuracy exceeded that of SEAS5 (left) and the linear trend (right) across the nine validation and test years (2012-2020), shown for each forecast calendar month and lead time. Since data from Oct 2020-Dec 2020 was not used, the maximum number of years is 8 for Oct-Dec.

Comment 7

Line 33-34: “It also demonstrates a greater ability to predict anomalous pan-Arctic sea ice extents than the models submitted to the Sea Ice Outlook programme: I don’t think this statement can hold well since it appears to me that the IceNet’s performance is very close to the overall performance of all models in the SIO (Fig.3). The SIO is a little better than IceNet in 2 and 4-month leads, actually.

Response 7

While the sea ice extent (SIE) metric adopted for the SIO model intercomparison framework is a standard metric, useful for assessing the large-scale changes in sea ice (e.g. interannual variability and impact of global warming), it suffers from several limitations for forecasting assessment. For example, the predicted sea ice edge can deviate severely from the observed one, while still having a low SIE error if the total area covered by sea ice is similar between the forecast and observation. The absolute SIE error is a lower bound on the *integrated ice edge error* (Goessling et al. 2016), which is a more appropriate metric for sea ice forecasting, measuring the total sea ice misclassification area. The integrated ice edge error can be seen as

an unnormalised version of our binary accuracy metric, and we have added a section to the Methods at Lines 464-470 explaining how the two metrics are related. Therefore, the results comparing with the SIO, while providing a useful indicator of IceNet's performance relative to other models, should only be seen as upper bounds on sea ice edge prediction skill.

Our claim mentioned by the reviewer refers to our year-wise decomposition figure of September absolute SIE errors, showing IceNet's seasonal prediction of the SIE was better than the SIO in the anomalous ice extent years (2012, 2013, 2020). However, given the aforementioned limitations of the SIE for assessing forecasts, we have produced an equivalent year-wise analysis of IceNet's relative seasonal September forecasting ability based on our binary accuracy metric (Rebuttal Fig. 3), which provides a measure of the ice edge error and is a more reliable assessment of forecast performance.

Given the limitations of the SIE error metric and the more reliable results presented in Rebuttal Fig. 3, we have chosen to move the SIO comparison figure to the Supplementary Information (Supplementary Fig. 6) and replace it with Rebuttal Fig. 3 in the place of Figure 4. In accordance with this change, we have included two paragraphs at Lines 167-179 to describe these results and explain how they validate our claims about IceNet's relative reliability and robustness for forecasting extreme pan-Arctic September events. We have also included sentences at Lines 188-192 to explain the caveats of the SIE metric. Finally, we have modified the sentence in the abstract mentioned by the reviewer to be based on our new results comparing IceNet with SEAS5 and the linear trend (Line 39).

Rebuttal Fig. 3 | IceNet’s improvement in binary accuracy relative to SEAS5 and the linear trend models for September forecasts at 4- to 2-month lead times, shown for the validation and test years (2012-2020).

Comment 8

The authors purposely compare IceNet with a single dynamical model (SEAS5). In my view, a more appropriate evaluation should be made to compare IceNet with all dynamical models participating in the SIO since it is unknown where SEAS5 lies in the spectrum of overall performance derived from all available dynamical models in the community.

Response 8

We agree with the reviewer that a rigorous comparison between IceNet and a suite of dynamical models would be a prudent evaluation of performance. However, the performance of SEAS5 relative to other dynamical models is well documented in the literature. We refer

the reviewer to Fig. 1 of Wayand et al. 2019, which shows that SEAS5 (labelled as ECMWF-C3S) has the most consistently low pan-Arctic Brier Score up to a 6-month lead time, compared with 12 other dynamical models. SEAS5 is even comparable to the multi-model ensemble of all the individual models, which is often a very strong benchmark. We also refer the reviewer to Zampieri et al. 2018, where Fig. 1 shows SEAS5 substantially outperforming other dynamical models at short lead times. Both of these studies are cited on Line 136 when we introduce SEAS5 as having ‘state-of-the-art sea ice prediction skill’.

We chose to only compare IceNet with the leading dynamical sea ice forecasting model in order to significantly simplify the narrative of the manuscript, which we believe makes it more enjoyable to read and digest, while not sacrificing technical validity.

Furthermore, accessing and preprocessing a single dynamical model’s forecasts can be an arduous task. Differing hindcast periods, data access methods, land-sea masks, and metadata can necessitate weeks of work. Therefore, including a whole suite of dynamical models in this study would substantially delay publication.

While it is not our intention to perform a full intermodel comparison in this introductory study, our held-out forecast data will be made publicly available upon publication to facilitate such future studies. Given IceNet’s strong performance in seasonal forecasts, we expect the publication of this study to naturally generate interest from sea ice forecasting researchers who wish to use IceNet’s predictions as a benchmark for comparisons with their models. We also note that, following discussions at a Sea Ice Prediction Network workshop, IceNet will be included in a future study comparing sea ice forecasting methods, and we hope the reviewer will find such a model intercomparison enlightening.

Reviewer 2 responses

Comment 1

While the choice of predictors may seem logical for their relation to changes in sea ice, some references here on why these variables were chosen should be included.

Response 1

In response to the reviewer's comment, which was also raised by R1, we have added a new section to the main text (Lines 118-132) entitled 'IceNet's input variables'. We have attempted to strike a balance between providing justification behind IceNet's input variables, while not diving too deeply into the complex topic of atmosphere-sea ice-ocean couplings. To allow the reader to better understand the drivers underlying sea ice mass balance, we have cited two books on sea ice physics and dynamics (alongside other references).

The inclusion of this new section has also allowed us to include a paragraph raising the point that not all variables relevant to sea ice changes are sufficiently observed to be used in IceNet (Lines 128-132), which highlights the fact that predictive capacity will be limited to some extent due to missing information.

We thank R2 and R1 for encouraging us to include this section, which has improved the flow of the paper. We believe this paragraph will benefit readers from the machine learning community, who may wish to improve upon IceNet in their own research, but lack the scientific background to understand the choice of input variables.

Comment 2

Table 1a and 1b show the drop in accuracy for the top 5 variables. The 5th ranked variables' drop in accuracy is roughly $\sim 0.2\%$. I am curious what lower ranked variables score? It seems many of the variables might be having quite a negligible (maybe even harmful) affect on the predictions. Did you use this method to prune variables that are not helpful? I would imagine this could improve computation costs or allow you to explore other variables that have proven to be influential on the evolution of sea ice (i.e., longwave radiation, specific humidity, vapor transport). As you mention, knowledge of which variables are influential for

sea ice forecasting could be useful for improving dynamic models, but knowledge of which variables have no influence could be of use as well.

Response 2

We thank the reviewer for posing this interesting question regarding the lower-ranked input variables. In response, we provide the full permute-and-predict results (averaged over all forecast months in 2012-2019) in Rebuttal Fig. 4, which we have included in the Supplementary Information (Supplementary Fig. 7). Although certain variables may play a greater or smaller role in the predictability of sea ice in different seasons, these averaged results highlight the permuted variables that result in the largest or smallest accuracy drops over the whole calendar year. Each mean accuracy drop shown is the mean over $N_{\text{seeds}} \times N_{\text{years}} \times N_{\text{calendar months}} = 10 \times 8 \times 12 = 960$ values.

Here we see that it is rare for a variable to cause a mean accuracy increase upon permutation, however, several variables have values close to zero. This alone does not imply that IceNet is not using those variables for its predictions however, because the spread around the mean accuracy drop could still be large. Furthermore, it is difficult to answer questions about which variables might be harmful for IceNet from these results alone. For that, a more appropriate variable importance method is the drop-and-relearn experiment described in Response 3 to R1, which we will include in a future study after inspiration from this round of peer review.

Rebuttal Fig. 4 | Mean accuracy change from the permute-and-predict method for each input variable and lead time. The number in brackets next to the variable names indicate the input lag (or for the linear trend forecast input, the lead) in months. The colorbar is artificially saturated at -0.5% to emphasise the patterns at smaller magnitudes. SIC = sea

ice concentration.

As well as highlighting variables with small importance values from permute-and-predict, Rebuttal Fig. 4 also reveals interesting new patterns of variables with large accuracy drops that are not apparent from the top-5 ranking in Table 1. For example, the reduced IceNet ensemble used for this experiment seems to be more sensitive to permutations of the 500 hPa geopotential height anomaly field than the 250 hPa field, despite the strong correlations between the two pointed out by the reviewers. This may be because lower altitude winds have a more dominant effect on sea ice and IceNet has identified this stronger relationship.

This new supplemental figure is the basis of an extended discussion on the permute-and-predict results on Lines 273-279 of the main text and Lines 775-777 of the Supplementary Methods.

Comment 3

How does this method handle collinearity among input variables? You mention that removing the seasonal cycle from non-SIC variables removes a lot of correlation, but I believe some variable anomalies would likely still be highly correlated (same variable at different pressure levels, air temperature and geopotential height...).

Response 3

We thank the reviewer for this prudent question and refer them to Comment 2 from our response to R1 where we address this concern.

Comment 4

Line 77: Why start with three classes then reduce to two? Why not just find the probability of ice/no ice? I might be misunderstanding this process, so either way I think it should be explained more clearly.

Response 4

We appreciate the feedback and have adjusted the text at Lines 94-102 to make this more clear, emphasising that the use of three classes was an early design choice to improve the expressivity of IceNet's forecasts. We have also moved the following sentence from the

model evaluation section to the section where the ice class boundaries are defined: ‘*By reducing the task to binary classification, the objective can be framed as that of predicting the ice edge.*’ To further highlight this point, we have moved the figure showing IceNet’s seasonal ice edge forecasts for Sept 2012, 2013 and 2020 (previously Figure 4) to the place of Figure 2 in the main text. We feel this also provides useful qualitative insights into IceNet’s performance before the quantitative analysis section that follows.

Although the use of three classes means IceNet is trained to predict both the sea ice edge and the inner edge of the marginal ice zone (SIC = 80%), the reviewer may be interested to learn that we also tried training IceNet as a binary classification model, and found this not to provide any performance benefits on the binary problem compared with the three-class version.

Comment 5

Line 95: How sensitive are the results to the choice of emission scenarios?

Response 5

Our ablation study showed that the CMIP6 pre-training phase did not have a significant effect on model performance, which is likely due to limitations in model physics (see Line 195). As such, we would not expect significant changes to our results based on the emission scenario alone, and it would be difficult to draw conclusions from such a sensitivity analysis. While we feel the substantial amount of work required to perform such an experiment would be more appropriate for a future study, in response to the reviewer’s question we compared the performance on the extreme September years (2012, 2013 and 2020) for the CMIP6 pre-trained IceNet ensemble and the ensemble without CMIP6 pre-training. We found no systematic difference in prediction of these extremes with CMIP6 pre-training, further suggesting that the sensitivity of our results to the choice of emission scenario is likely to be small.

Comment 6

Line 106: Use a linear trend of what? Sea ice concentration? How do you get a probabilistic forecast from this for comparison to your model’s output?

Response 6

The linear trend produces deterministic SIC forecasts only. As mentioned in Line 145, its output SIC is converted to binary predictions based on the 15% SIC threshold. It is only used as a benchmark for forecast accuracy. Therefore, it does not appear in Figure 7 or Supplementary Figures 1-2, where we compare IceNet's probabilistic calibration with SEAS5. However, we agree the sentence mentioned by the reviewer could be more clear and have adjusted it to read 'SIC linear trend forecast' at Line 140.

Comment 7

Line 109: Is it not problematic to use the linear trend as a predictor and then compare your output to the linear trend? Wouldn't your model necessarily outperform the linear trend in this case?

Response 7

We agree with the reviewer that, in most cases, one would expect IceNet to outperform models whose forecasts are passed in as inputs. However, IceNet may only be able to match the performance of those forecasts if it cannot learn to leverage the other inputs to improve performance further. Thus, IceNet's performance relative to the linear trend model provides valuable information on which forecasting regimes IceNet can use the 'initial conditions' variables to improve upon the simple linear trend forecasts. Inputting the linear trend forecasts therefore provides an additional layer of interpretability with IceNet, as well as improving performance. We have added a sentence at Lines 141-143 to emphasise this point and thank the reviewer for helping us to improve the clarity on this.

Comment 8

Lines 342-343: Remove "The active grid cell area s."

Response 8

We appreciate the reviewer's careful proofreading. We have removed this error from the manuscript at Line 413.

Comment 9

Line 642: Did you look at the distribution of the accuracy drop values? Are they normally distributed? Are there outliers? Would the median work just as well as the mean?

Response 9

We thank the reviewer for this interesting comment regarding the distribution of accuracy drop values. We note that it is standard practice to report only the mean drop (Breiman 2001, Section 10). As the intention in this study was simply to use an existing machine learning interpretability method rather than modify or extend it, it would not be appropriate to make changes to the manuscript to explore the distribution and use of the median. However, in response to this comment, we have produced some example histograms of the accuracy drop values for the interest of the reviewer. As noted by the reviewer and stated on Line 763 in the Supplementary Methods, each accuracy drop in Table 1a and Table 1b is the mean of 80 values (10 random seeds x 8 years). Rebuttal Fig. 5 shows the accuracy drop histograms from permuting the initialisation SIC field for September forecasts. For reference, the 1-month lead time histogram from Rebuttal Fig. 5 shows the values used for the mean accuracy drop given in the first row and third column of Table 1a.

The spread in the histograms are due partly to the random permutation seed (the perturbation to the input will depend on the data it is replaced by), and partly to the changes between years. The distributions sometimes look like a normal distribution with truncated tails and sometimes like a uniform distribution, or perhaps some mixture of the two. There is also a notable spike in values close to 0 on some occasions (corresponding to small accuracy change due to permutation). As the histograms appear to have little skewness, we would expect the median and the mean accuracy drop values to be similar. However, the mean may be more desirable because it responds to outliers that may signify greater variable importance.

Permute-and-predict accuracy drops
Target month: September.
Variable: Initialisation SIC

Rebuttal Fig. 5 | Histogram of the 80 permute-and-predict accuracy drop values for September forecasts when the initialisation SIC input is permuted, shown for all 6 lead times. SIC = sea ice concentration.

Reviewer 3 responses

Comment 1

Still, the reviewer may wonder is there any further conclusions can be drawn from this paper/presented results in the aspect of environment or climate changes? Or other insights beyond the technical highlights.

Response 1

We thank the reviewer for their kind evaluation of our paper. We believe the technical highlights of our study already place it as a significant work within the sea ice forecasting literature, for example: outperforming a state-of-the-art dynamical model in seasonal forecasts while running 1,000s of times faster; improved fidelity in forecasting extreme sea ice events; and a unique ability to probabilistically bound the sea ice edge (with clear implications for Arctic shipping). Despite these highlights, we strongly agree that it is important to discuss insights other than the technical. However, we feel that our Discussion section already goes beyond that of a typical forecasting system paper. In particular, to our knowledge, our manuscript is the first to discuss the potential use of sea ice forecasts to aid conservation work in the Arctic and define dynamic marine protected areas (Lines 296-313). As stated on Line 305, *“Such applications would help to fill an urgent gap in the integration of climate change in ecosystem management and planning tools”*. Our results come at a crucial point in the decline of Arctic sea ice. Paragraph C.4.5 of the IPCC Special report on the Oceans and Cryosphere in a Changing Climate's Summary for Policy Makers states that, *“forecasting of changes in the ocean and the cryosphere informs adaptation planning and implementation”*. Despite this, little work has been done to strategically utilise sea ice forecasts for mitigating the multitude of ecological and social consequences of sea ice decline. This could in part be due to the low skill of dynamical models, which has historically limited their adoption for planning & adaptation. IceNet represents a step-change advancement in sea ice forecasting skill, which we believe will bring new attention to sea ice forecasts from groups who may otherwise have low expectations of forecast skill, as well as continued advancements from further development of our AI method.

The reviewer may be interested to know that, since our initial submission of the manuscript, we have developed a daily version of the model (IceNet2) and we have secured funding to

run the model in real-time as the first public, operational sea ice forecasting AI. Furthermore, we are collaborating with the WWF to explore how sea ice forecasts can be used for mitigating and adapting to environmental risks associated with sea ice loss. Our collaboration with WWF will lay the groundwork for the use of sea ice forecasts as an early warning system, giving decision-makers time to adapt to extreme events.

We would also like to highlight the discussion beginning on Line 260 on how analysis of our variable sensitivity results can provide evidence for the predictability of sea ice, particularly the sentence *“the predictability that IceNet can learn from training data provides evidence for the inherent timescales of memory in the Arctic climate system”*. In response to R2, this discussion has been extended with a new paragraph on Lines 273-279 and the addition of our full permute-and-predict results in Supplementary Figure 7. We hope the reviewer agrees that this analysis provides further insights on the drivers of sea ice changes beyond forecasting performance alone.

Comment 2

L65-66 The authors mention sea ice sensing using CNN. It may be more appropriate to include associated work as refs. such as Q. Yan and W. Huang, “Sea ice sensing from GNSS-R data using convolutional neural networks,” *IEEE Geosci. Remote Sens. Lett.*, vol. 15, no. 10, pp. 1510–1514, Oct. 2018. and L. Wang, K. A. Scott, L. Xu, and D. A. Clausi, “Sea ice concentration estimation during melt from dual-pol SAR scenes using deep convolutional neural networks: A case study,” *IEEE Trans. Geosci. Remote Sens.*, vol. 54, no. 8, pp. 4524–4533, Aug. 2016.

Response 2

We thank the reviewer for suggesting we cite relevant studies that apply CNNs to sea ice data. We have included the citations when the CNN model is first introduced at Line 71.

Comment 3

L106-110 Is there any selection of forecast to be passed into the IceNet based on the performance of the corresponding linear trend model?

Response 3

We may not be correctly understanding the reviewer's question, but we only use a single linear trend model in this study, which extrapolates the lines of best fit of SIC at each grid cell (see Line 137), and use this model as a statistical benchmark. The reviewer may be interested in Response 7 to R2 for further reasoning behind our inclusion of the linear trend model's forecasts as input to IceNet. We hope this clears up any concerns.

Rebuttal references

Belkin, M., Hsu, D., Ma, S. & Mandal, S. Reconciling modern machine-learning practice and the classical bias–variance trade-off. *PNAS* 116, 15849–15854 (2019).

Breiman, L. Random Forests. *Machine Learning* 45, 5–32 (2001).

Cavalieri, D. J., C. L. Parkinson, P. Gloersen, and H. J. Zwally. 1996, updated yearly. Sea Ice Concentrations from Nimbus-7 SMMR and DMSP SSM/I-SSMIS Passive Microwave Data, Version 1. Boulder, Colorado USA. NASA National Snow and Ice Data Center Distributed Active Archive Center. doi: <https://doi.org/10.5067/8GQ8LZQVL0VL>. [Accessed 5/5/2021].

De Veaux, R. D. & Ungar, L. H. Multicollinearity: A tale of two nonparametric regressions. in *Selecting Models from Data* (eds. Cheeseman, P. & Oldford, R. W.) 393–402 (Springer, 1994). doi:10.1007/978-1-4612-2660-4_40.

Goessling, H. F., Tietsche, S., Day, J. J., Hawkins, E. & Jung, T. Predictability of the Arctic sea ice edge. *Geophysical Research Letters* 43, 1642–1650 (2016).

Guemas, V. et al. A review on Arctic sea-ice predictability and prediction on seasonal to decadal time-scales. *Quarterly Journal of the Royal Meteorological Society* 142, 546–561 (2016).

Hooker, G. & Mentch, L. Please Stop Permuting Features: An Explanation and Alternatives. *arXiv e-prints* 1905, arXiv:1905.03151 (2019).

IPCC, 2019: Summary for Policymakers. In: *IPCC Special Report on the Ocean and Cryosphere in a Changing Climate* [H.-O. Pörtner, D.C. Roberts, V. Masson-Delmotte, P. Zhai, M. Tignor, E. Poloczanska, K. Mintenbeck, A. Alegría, M. Nicolai, A. Okem, J. Petzold, B. Rama, N.M. Weyer (eds.)]. In press.

Johnson, M. et al. Evaluation of Arctic sea ice thickness simulated by Arctic Ocean Model Intercomparison Project models. *Journal of Geophysical Research: Oceans* 117, (2012)

Wayand, N. E., Bitz, C. M. & Blanchard-Wrigglesworth, E. A Year-Round Subseasonal-to-Seasonal Sea Ice Prediction Portal. *Geophysical Research Letters* 10.

Zampieri, L., Goessling, H. F. & Jung, T. Bright Prospects for Arctic Sea Ice Prediction on Subseasonal Time Scales. *Geophys. Res. Lett.* 45, 9731–9738 (2018).

REVIEWERS' COMMENTS

Reviewer #1 (Remarks to the Author):

The authors have done a lot of extra work on this manuscript to address all concerns I brought up in the first round. I find the paper acceptable at this stage, and thank the authors for taking the time to respond so thoroughly.

Reviewer #2 (Remarks to the Author):

The authors have addressed all of my concerns. I recommend this manuscript for publication.

Reviewer #3 (Remarks to the Author):

The authors addressed all of my comments, I have no more questions.

Reviewer comments (second round)

Review #1

The authors have done a lot of extra work on this manuscript to address all concerns I brought up in the first round. I find the paper acceptable at this stage, and thank the authors for taking the time to respond so thoroughly.

Review #2

The authors have addressed all of my concerns. I recommend this manuscript for publication.

Review #3

The authors addressed all of my comments, I have no more questions.

Responses to reviews

We thank the reviewers again for helping us to improve the manuscript and for assessing our response to the first round of reviews. We are grateful for the unanimous recommendation of publication at this stage.